# Towards Universal Video Retrieval: Generalizing Video Embedding via Synthesized Multimodal Pyramid Curriculum

## Abstract

The prevailing video retrieval paradigm is structurally misaligned, as narrow benchmarks incentivize correspondingly limited data and single-task training. Therefore, universal capability is suppressed due to the absence of a diagnostic evaluation that defines and demands multi-dimensional generalization. To break this cycle, we introduce a framework built on the co-design of evaluation, data, and modeling. First, we establish the Universal Video Retrieval Benchmark (UVRB), a suite of 16 datasets designed not only to measure performance but also to diagnose critical capability gaps across tasks and domains. Second, guided by UVRB's diagnostics, we introduce a scalable synthesis workflow that generates 1.55 million high-quality pairs to populate the semantic space required for universality. Finally, we devise the Modality Pyramid, a curriculum that trains our General Video Embedder (GVE) by explicitly leveraging the latent interconnections within our diverse data. Extensive experiments show GVE achieves state-of-the-art zero-shot generalization on UVRB. In particular, our analysis reveals that popular benchmarks are poor predictors of general ability and that partially relevant retrieval is a dominant but overlooked scenario. Overall, our co-designed framework provides a practical path to escape the limited scope and advance toward truly universal video retrieval.

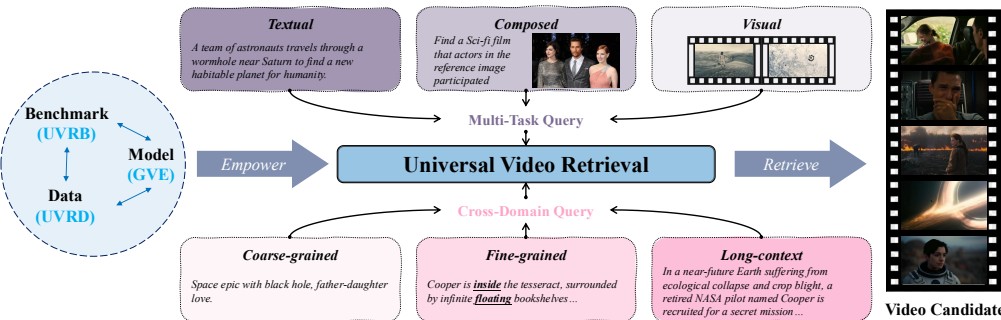

Figure 1: We propose Universal Video Retrieval (UVR) that retrieves videos with multi-task, cross-domain queries, which can be achieved via benchmark-data-model co-design in this work.

## 1 Introduction

Video retrieval is a critical, yet challenging task for modern search engines and recommendation systems, requiring effective video embedding models (Zhu et al., 2023). Early efforts extended Contrastive Language-Image Pretraining (CLIP) (Radford et al., 2021) to video (Ma et al., 2022). Now, a paradigm shift is underway, with Multimodal Large Language Models (MLLMs) rapidly displacing CLIP for their superior language understanding and visual generalization capabilities (Kong et al., 2025). Current practice involves training these models on massive datasets with simple and noisy text annotations (e.g., WebVid (Bain et al., 2021)) with strong results for coarse-grained text-to-video retrieval on benchmarks (e.g., MSRVTT (Xu et al., 2016)).

However, they struggle with the complexity of diverse video retrieval scenarios (Figure 1). First, a narrow semantic distribution renders these models ineffective in fine-grained queries required to

understand spatial relations or temporal dynamics (Xu et al., 2025), as well as in long-context retrieval within lengthy videos (Cai et al., 2025). Second, the scope of applicable tasks is restricted, with little support for diverse query formats beyond plain text, such as composed retrieval using text-and-image pairs and purely visual queries. Existing specialized models (Hummel et al., 2024) are costly and hinder progress toward a single, generalizable model across these emerging scenarios.

Therefore, to establish a framework for universal video retrieval that supports multi-domain, multi-granularity, and multi-task capabilities, three coupled challenges in evaluation, data, and modeling need to be addressed simultaneously: (1) *Dimensional Diagnostic Evaluation*: The foundational step is to define and measure universality quantitatively. This necessitates a comprehensive evaluation framework capable not only of assessing performance across diverse tasks but, more critically, of diagnosing the intricate correlations and interferences between them. (2) *Large-Scale Quality-Controlled Data Synthesis*: Existing datasets are either too small or are biased, while collecting a new, massive dataset is prohibitively expensive. Growing works attempt to synthesize data to address this challenge (Chen et al., 2023; Chai et al., 2024; Ventura et al., 2024). However, these resources often exhibit uneven quality and distribution. Therefore, gaining precise control over the properties of large-scale, cross-domain, multi-task data via a unified synthesis process is the second challenge. (3) *Interconnected Multi-task Representation Learning*: A critical, often-ignored aspect is the inherent hierarchy among tasks. Foundational abilities such as spa-

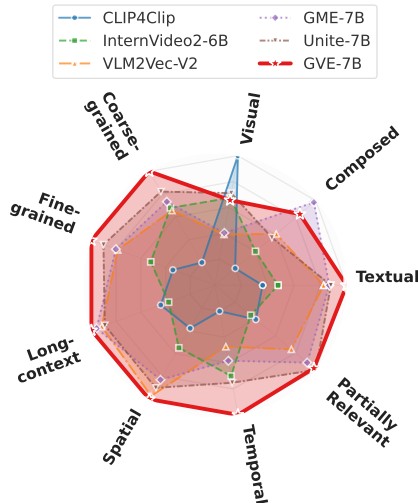

Figure 2: Model performance on UVRB for 9 abilities (3 main tasks and 6 (sub-) domains).

tial perception (e.g., object recognition) serve as building blocks for higher-order temporal reasoning (e.g., action recognition). This principle is evidenced by the remarkable success of image-only trained models like GME (Zhang et al., 2025) on video retrieval tasks (see Figure 2), highlighting latent cross-task adaptability. Nevertheless, conventional models that assume task independence fail to capitalize on this structure, which is a key to unlocking superior generalization.

To this end, we propose a holistic framework by co-designing evaluation, data, and modeling. Specifically, we first construct the **Universal Video Retrieval Benchmark (UVRB)**, a comprehensive suite with 16 test datasets across diverse domains and tasks for 14 state-of-the-art models. More importantly, an in-depth analysis quantitatively exposes the limitations of current approaches. Second, based on the diagnostics, we design **V-SynFlow**, a multi-stage data synthesis workflow that transforms massive, low-quality text-video pairs into a high-quality and multi-task dataset, **Universal Video Retrieval Dataset (UVRD)**. It consists of over 1.55 million video retrieval pairs with rich spatial-temporal details, diverse descriptive styles, and distinct task formats. Third, we devise the MODALITY PYRAMID, a customized curriculum that leverages inherent task and domain dependencies to optimize a **General Video Embedder (GVE)** on the diverse synthesized data for advanced zero-shot task and domain adaptation. The bottom-up, pyramid-shaped curriculum prioritizes data-abundant, foundational tasks before progressing to more complex, dependent ones for progressive and stable knowledge acquisition. Extensive experiments on UVRB validate the effectiveness of GVE (Figure 2). Besides, the diagnostic analysis reveals underexplored findings. For example, conventional benchmarks are not representative of the overall retrieval ability, indicating the potential for the overfitting of existing models on in-domain data. Instead, partially relevant video retrieval, despite low research attention, is a typical and generalizable scenario in this field.

The contributions of this work are summarized as follows: **(1) Benchmark**: A universal video retrieval benchmark with 16 test datasets for the multi-dimensional, diagnostic capability evaluation. **(2) Data**: A scalable video data synthesis workflow, producing over 1.55 million cross-domain and multi-task pairs to establish a high-quality training resource. **(3) Training & Model**: A multimodal pyramid curriculum for learning generalizable video embeddings by modeling inherent knowledge dependencies across tasks. **(4) Experiment & Analysis**: Extensive experimental results and analysis, validating the superiority of our proposed methods among 14 state-of-the-art video retrievers and discovering unnoticed and insightful knowledge.

## 2 RELATED WORKS

**Video Retrieval.** Text-to-video retrieval has progressed from matching coarse phrases to parsing fine-grained spatio-temporal descriptions (Xu et al., 2016). While recent benchmarks have advanced beyond simple recognition by incorporating detailed annotations like spatio-temporal grounding (CaReBench (Xu et al., 2025)), scene understanding (UltraVideo (Xue et al., 2025)), and camera motion (CameraBench (Lin et al., 2025)), they remain specialized. Concurrently, the scope of retrieval has expanded to include new paradigms like composed queries with text-image pairs (e.g., CoVR (Ventura et al., 2024; Hummel et al., 2024)) and purely visual queries in egocentric contexts (Liu et al., 2021). Despite these advances, the field remains fragmented, with models and evaluation siloed within specific tasks or domains. This prevents a holistic understanding of a model's true generalization capabilities, a gap our unified benchmark aims to fill.

**Video Embedding Models.** Video embedding models have evolved from the adaptation of image-centric ones, such as CLIP (Radford et al., 2021), to powerful, larger language-based encoders. Early methods such as CLIP4Clip (Luo et al., 2022) and InternVideo2 (Wang et al., 2024c) added temporal modules to CLIP but inherited its limitations in complex language understanding and long-context processing (Wang et al., 2025; Li et al., 2025). To overcome these issues, recent work leverages Multimodal Large Language Models (MLLMs) as video embedders. Models like LLaVE (Lan et al., 2025), UNITE (Kong et al., 2025), and VLM2Vec-V2 (Meng et al., 2025) achieve strong performance on benchmarks by training on text-image and text-video data. However, this data has so far been narrow for learning generalizable embeddings, and these video embedding models have not been systematically evaluated across more complex tasks and domains of video retrieval.

## 3 METHODOLOGY

This section establishes a new ecosystem to reshape the fragmented scope of video retrieval by a co-designed, tripartite framework. The basis is the **Universal Video Retrieval Benchmark (UVRB)**, which defines a comprehensive suite of abilities and serves as a diagnostic tool. Informed by UVRB's diagnostics, our **V-SynFlow** pipeline synthetically generates a high-fidelity dataset, **Universal Video Retrieval Dataset (UVRD)**, engineered to populate the identified semantic and structural gaps. Finally, the **MODALITY PYRAMID** provides a principled curriculum with adaptive task scheduling to train a **General Video Embedder (GVE)**. This tight integration of diagnostic evaluation, targeted data synthesis, and model optimization forms a feasible solution for universal video retrieval.

### 3.1 UNIFYING AND BENCHMARKING VIDEO RETRIEVAL

Existing works are typically confined to coarse-grained text-to-video tasks, limiting their capacity to define and evaluate model generalization. To address the critical flaw in video retrieval evaluation, we propose the paradigm of **Universal Video Retrieval (UVR)**, defined below.

**Definition 1** (Universal Video Retrieval (UVR))**.** *Given a related pair with a query $q$ and a video $v$, UVR aims to learn a $\theta$-parameterized model $E_\theta(\cdot)$ to compute a relevance score between their embeddings, $s_{q \to v} = \cos(E_\theta(q), E_\theta(v))$, which should be higher than other irrelevant pairs. This condition can be satisfied for the given pair with different formats and in divergent domains. Specifically, (1) Query Format can be Textual (TXT, e.g., natural language), Composed (CMP, text+image/text+video), Visual (VIS, image/video). (2) Data Domain can be Coarse-grained (CG, high-level semantics), Fine-grained — Spatial (S, object appearance), Temporal (T, event dynamics), Partially Relevant (PR, local or abstract information) — and Long-context (LC, extended inputs).*

To enable a systematic evaluation of UVR, we introduce the **Universal Video Retrieval Benchmark (UVRB)** as shown in Figure 3. While UVRB is constructed from publicly available data, its scientific contribution is not data novelty but the establishment of a comprehensive diagnostic framework. This framework transcends simple dataset aggregation through three key contributions that address the fragmented nature of prior evaluation efforts.

First, inspired by seminal works in retrieval fields like MTEB (Muennighoff et al., 2022) and MMEB-V2 (Meng et al., 2025), UVRB establishes the field's first *principled capability taxonomy* for video retrieval. According to Definition 1, UVRB organizes the evaluation along two orthogonal axes:

Figure 3: The dataset category of Universal Video Retrieval Benchmark (UVRB).

query format and domain complexity. This transforms evaluation from a series of disconnected performance scores into a holistic diagnostic tool that reveals a model's strengths, weaknesses, and capability trade-offs across a unified semantic space.

Second, we undertook substantial transformative data curation to programmatically instantiate the full spectrum of our taxonomy, creating new retrieval paradigms where none existed. This effort included: (1) *Repurposing Annotations*: We converted annotations not intended for retrieval into novel queries, such as using detailed camera motion descriptions from CMRB (Lin et al., 2025) to create a temporal reasoning task. (2) *Reformatting Tasks*: We reconfigured datasets like PEV-K (Bolya et al., 2025), using its keyword lists for a partially-relevant matching task, a significant departure from its original descriptive format. (3) *Visual Processing*: We manipulate raw video data, such as sampling single frames from MSRVTT (Xu et al., 2016) to create the MSRVTT-I2V task. This curation process resulted in 16 distinct datasets targeting specific abilities[1]. Full details are in Appendix A.2.

Third, UVRB introduces a rigorous and unified evaluation protocol to ensure fair comparison. This protocol creates a controlled environment by standardizing three key aspects of the evaluation pipeline: (1) *Input Standardization*: Enforces consistent data presentation to all models, including a vision-only modality scope, a fixed video frame sampling strategy, and uniform limits on sequence length and visual token budgets. (2) *Model-Agnostic Representation Extraction*: Mandates that evaluation targets the core embedding space by prohibiting any model-specific post-processing, re-ranking, or the use of learnable projection heads. (3) *Uniform Scoring Mechanism*: Employs a consistent universal relevance metric (i.e., cosine similarity) for all tasks and models.

In summary, UVRB is a complete evaluation ecosystem beyond a dataset collection, designed for diagnostic insight. The insights from our evaluation of baselines (Section 4.2) and abilities (Section 4.3) directly inform and validate our subsequent contributions in data synthesis (Section 3.2) and model training (Section 3.3), embodying our co-design philosophy.

## 3.2 Scalable Synthesis of Cross-Domain Multi-Task Video Retrieval Data

Training a universal video embedder is fundamentally impeded by a deficiency of high-quality supervision for divergent tasks. Therefore, we introduce **V-SynFlow** (Figure 4) to transform weakly annotated web videos from raw datasets (e.g., PVD (Bolya et al., 2025), InternVid-FLT (Wang et al., 2023), and WebVid (Bain et al., 2021)) into a structured, high-fidelity, multi-task training instances. In this way, we obtain a practical and diverse dataset, called **Universal Video Retrieval Dataset (UVRD)** with over 1.55 million descriptive pairs in total for dimensional training enhancement (see Appendix A.4 for details). V-SynFlow proceeds in three stages: We first construct a clean, semantically coherent material pool by filtering noise at multiple granularities. Then we leverage an MLLM as a conditional generative engine to enrich semantic dimensions. Lastly, we synthesize diverse instances across multiple retrieval tasks. We provide our details and applied prompts for synthesis in Appendix A.6.

**Multi-granular Quality Control.** Given a raw corpus $\mathcal{D} = \{(v_i, t_i)\}$, we produce a high-fidelity asset pool, $\mathcal{A}_{tfc}$. The process applies a filter cascade: *Annotation Rectification* to remove non-descriptive text; *Cross-Modal Consistency Filtering*, which discards pairs where the similarity from a pretrained embedder $\Phi(\cdot)$ (e.g., GME-7B (Zhang et al., 2025)) is below a threshold; and *Temporal Dynamics Filtering* to remove static content. The resulting asset pool $\mathcal{A}_{tfc}$ contains a set of validated

---

[1]We define an **ability** as proficiency in one query format *or* one data domain (e.g., VIS is an ability; T is an ability). A general embedding model should master multiple abilities and their combinations.

Figure 4: V-SynFlow: a multi-stage synthesis workflow for diverse video retrieval data.

videos $\{v_j\}$, their original captions $\{t_j\}$, and corresponding sets of extracted frames $\{f_{jk}\}$ and cropped clips $\{c_{jl}\}$.

**Multi-dimensional Information Enrichment.** We leverage the filtered assets in $\mathcal{A}_{tfc}$ to generate richer data structures. To create an enriched text-video dataset $\mathcal{D}^+$, we use an MLLM, $\mathcal{M}_{cap}$ (e.g., Keye-VL-8B (Team et al., 2025)), as a conditional captioning engine. For each video $v_j$, it synthesizes multiple captions $\{t'_{jk}\}$ conditioned by randomly generated information profiles (30% spatial, 60% temporal, and 10% others). By sampling one of the captions for each video, we obtain a set of new high-quality text-video pairs $\{(v_j, t'_{jk})\}$. Besides, we form pairs of a video and its visual components to construct collections of visual pairs $\mathcal{P}_{f\leftrightarrow v}$ and $\mathcal{P}_{c\leftrightarrow v}$, resulting in frame-to-video $\{(f_{jk}, v_j)\}$ and clip-to-video $\{(c_{jl}, v_j)\}$ pairs, respectively.

**Multimodal Task Extension.** The final stage assembles the unified training corpus, $\mathcal{D}_\star$. We synthesize complex composed retrieval tasks by leveraging $\mathcal{P}_{f\leftrightarrow v}$ and $\mathcal{P}_{c\leftrightarrow v}$. For each visual pair (e.g., $(f_{jk}, v_j)$, $(c_{jl}, v_j)$), $\mathcal{M}_{cap}$ generates a query text $t^{f_{jk}\rightarrow v_j}$ describing the temporal evolution, forming a training instance $((t^{f_{jk}\rightarrow v_j}, f_{jk}), v_j)$. Besides, basic alignment tasks (e.g., text-image pairs) are also sampled from $\mathcal{A}_{tfc}$. For each video, two candidates from the unselected synthesized captions are mapped as text-to-text pairs. The resulting dataset $\mathcal{D}_\star$ provides a comprehensive mix of tasks essential for training a universal embedder.

**Human Validation.** To validate the quality of UVRD generated from this automated pipeline, a human expert evaluation confirmed a 95% factual accuracy rate on 100 random samples from our synthesized data (see Appendix A.13 for details).

### 3.3 MODALITY PYRAMID: CUSTOMIZED CURRICULUM CONTRASTIVE LEARNING FOR GENERALIZABLE EMBEDDINGS

Our synthesized dataset provides a rich mixture of retrieval tasks spanning diverse query formats (text, image, video, and their compositions) and data domains (coarse- to fine-grained, long-context, etc., see Appendix A.4 for details). To embed multimodal inputs into a unified space, we introduce **General Video Embedder (GVE)** (Figure 5), a multimodal encoder derived from Qwen2.5-VL (Bai et al., 2025) to inherit its pretrained vision-language aligned knowledge. For arbitrary modality combinations (e.g., image-only or text+video), GVE fuses the tokenized prompt and text inputs with projected visual features into a joint input sequence. The LLM processes this sequence to autoregressively produce representations. We extract the final embedding via last token pooling and $\ell_2$-normalization for retrieval. Details are provided in Appendix A.7.

However, our diagnostics based on UVRB reveal that naively training a single embedder on this heterogeneous data leads to suboptimal performance (Section 4.2). One of the potential reasons is that easy tasks dominate early optimization, while challenging ones receive insufficient gradient signal and converge poorly. Moreover, existing methods either train on single-task data or overlook the knowledge dependencies between heterogeneous domains of multi-task data, which can benefit the joint incorporation of model abilities. To address this, we propose **MODALITY PYRAMID** (Figure 6), a curriculum that schedules training from atomic to composite tasks for progressive knowledge acquisition. This bottom-up design is grounded in empirical evidence: after one epoch, we observed a clear task hierarchy based on alignment scores, descending from text-image (0.65) down to text-video (0.46) and composed video (0.43) tasks (see Appendix A.14 for full analytical results). The

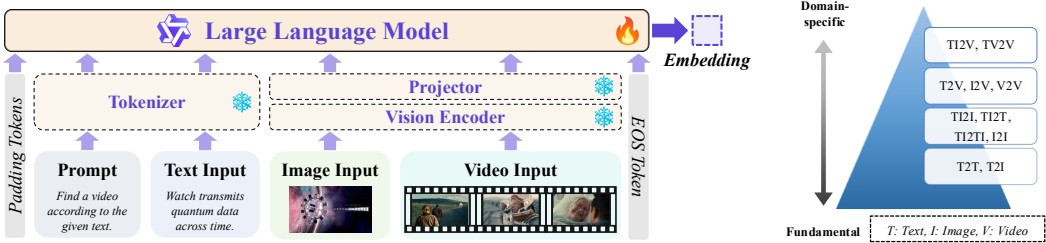

Figure 5: The architecture of GVE, a MLLM-based embedding model. We only fine-tune the LLM part. GVE inputs compositional multimodal elements and outputs a high-dimensional vector as an embedding.

Figure 6: MODALITY PYRA-MID: simpler tasks lay the foundation for specific ones.

curriculum guides the embedding model to master perceptual primitives first, then advance to complex integration, preserving foundational knowledge while cultivating generalizable transfer.

**Alignment-aware dynamic scheduling.** At the beginning of each training epoch $t$, we estimate the alignment level of every task $k \in \mathcal{K}$ using a *prober model* $\Psi_t$. For $t = 1$, $\Psi_1$ is a strong off-the-shelf embedder (e.g., GME-7B (Zhang et al., 2025)). The choice of a strong, MLLM-based prober is deliberate to ground our curriculum in reliable, semantically-rich task difficulty estimates, as a biased or weaker prober could mis-order the learning sequence. For $t > 1$, $\Psi_t$ is the GVE checkpoint from the end of epoch $t-1$. For each task, we sample $N_p$ positive pairs and compute its relevance score as the average cosine similarity: $R_k^{(t)} = \frac{1}{N_p} \sum_i \cos(\Psi_t(x_i), \Psi_t(y_i))$. Higher $R_k^{(t)}$ indicates better current alignment. During epoch $t$, tasks are sampled with probability $P^{(t)}(k) \propto \exp(R_k^{(t)}/\sigma(t))$, where the temperature $\sigma(t)$ increases linearly from $\sigma_{\min}$ to $\sigma_{\max}$ ($\sigma_{\min} = 0.1$ and $\sigma_{\max} = 1.0$ by default). This annealing schedule ensures initial focus on well-aligned tasks while progressively encouraging exploration of challenging ones.

**Unified Contrastive Optimization.** GVE is trained with a symmetric InfoNCE loss (Oord et al., 2018) across all scheduled tasks. To strengthen discrimination, we augment in-batch negatives with hard negatives mined from a large external corpus using the same prober $\Psi_t$. For a query–target pair $(q, v^+)$, the loss contrasts $v^+$ against both other positives in the batch and the top-$K$ retrieved hard negatives. The similarity is computed as $s_{q \to v} = \cos(E_\theta(q), E_\theta(v))$, and the final loss is symmetrized over query and target directions:

$$\mathcal{L}_i^{(q \to y)} = -\log \frac{\exp(s_{q_i \to y_i^+}/\tau_l)}{\exp(s_{q_i \to y_i^+}/\tau_l) + \sum_{j \neq i} \exp(s_{q_i \to y_j^+}/\tau_l) + \sum_{y_k^- \in \mathcal{H}_i} \exp(s_{q_i \to y_k^-}/\tau_l)}, \quad (1)$$

where $s_{q \to v} = \cos(E_\theta(q), E_\theta(v))$ and $\tau_l$ is a pre-defined temperature. The total loss is the symmetric sum $\mathcal{L}_i = \frac{1}{2}(\mathcal{L}_i^{(q \to y)} + \mathcal{L}_i^{(y \to q)})$.

## 4 EXPERIMENTS

### 4.1 EXPERIMENTAL SETUPS

**Baselines.** Our evaluation benchmarks 14 prominent baselines. As detailed in Appendix A.9, our selection spans a wide range of architectures, parameter sizes (from 87M to 8.3B), and training data compositions. The models are broadly divided into two categories: First, traditional *CLIP-based* embedding models include `CLIP4Clip` (Luo et al., 2022), `ViCLIP` (Wang et al., 2023), `VideoCLIP-XL` (Wang et al., 2024a), `LanguageBind` (Zhu et al., 2024), and the `InternVideo2` series (1B and 6B) (Wang et al., 2024c). Second, a more recent category of *MLLM-based* embedding models includes `GME-2B` (Zhang et al., 2025), `Unite-2B` (Kong et al., 2025), `VLM2Vec-V2` (Meng et al., 2025), `BGE-VL` (Zhou et al., 2024), `UniME-7B` (Gu et al., 2025), `B3-7B` (Thirukovalluru et al., 2025), `GME-7B` (Zhang et al., 2025), and `Unite-7B` (Kong et al., 2025). Note that the training data of baseline models may include in-domain data of test datasets in UVRB (e.g, MSRVTT, DiDeMo).

Table 1: Video retrieval performance for datasets of UVRB. The AVG values are averaged over 16 datasets. For each column: highest score is **bolded**, second-highest is underlined. Metrics: R@1 (Recall@1), R@10 (Recall@10), P@1 (Precision@1).

| Model | AVG | MSRVTT | DiDeMo | CRB-G | CRB-S | VDC-O | CRB-T | CMRB | DREAM-E |
|---|---|---|---|---|---|---|---|---|---|
| | | R@1 | R@1 | R@1 | R@1 | R@1 | R@1 | R@10 | R@1 |
| CLIP4Clip | 0.390 | 0.333 | 0.297 | 0.511 | 0.497 | 0.620 | 0.289 | 0.280 | 0.191 |
| ViCLIP | 0.352 | 0.386 | 0.306 | 0.447 | 0.437 | 0.530 | 0.349 | 0.229 | 0.235 |
| VideoCLIP-XL | 0.491 | 0.443 | 0.403 | 0.828 | 0.839 | 0.735 | 0.487 | 0.274 | 0.263 |
| LanguageBind | 0.487 | 0.479 | 0.421 | 0.716 | 0.687 | 0.759 | 0.466 | 0.290 | 0.280 |
| InternVideo2-1B | 0.404 | 0.449 | 0.404 | 0.586 | 0.568 | 0.644 | 0.470 | 0.355 | 0.242 |
| InternVideo2-6B | 0.427 | **0.485** | 0.418 | 0.608 | 0.612 | 0.650 | 0.455 | 0.346 | 0.271 |
| GME-2B | 0.488 | 0.390 | 0.303 | 0.690 | 0.718 | 0.715 | 0.400 | 0.298 | 0.240 |
| Unite-2B | 0.480 | 0.367 | 0.298 | 0.699 | 0.723 | 0.727 | 0.409 | 0.284 | 0.223 |
| VLM2Vec-V2 | 0.508 | 0.330 | 0.299 | 0.828 | 0.843 | 0.775 | 0.410 | 0.286 | 0.228 |
| BGE-VL | 0.443 | 0.337 | 0.318 | 0.690 | 0.688 | 0.639 | 0.359 | 0.225 | 0.212 |
| UniME-7B | 0.521 | 0.351 | 0.335 | 0.815 | 0.827 | 0.743 | 0.476 | 0.317 | 0.293 |
| B3-7B | 0.511 | 0.282 | 0.350 | 0.815 | 0.825 | 0.768 | 0.415 | 0.312 | 0.216 |
| GME-7B | 0.530 | 0.436 | 0.377 | 0.740 | 0.767 | 0.731 | 0.442 | 0.304 | 0.274 |
| Unite-7B | 0.538 | 0.439 | 0.386 | 0.798 | 0.804 | 0.753 | 0.472 | 0.351 | 0.279 |
| GVE-3B | 0.544 | 0.431 | 0.376 | 0.850 | 0.846 | 0.786 | 0.496 | 0.363 | 0.280 |
| GVE-7B | **0.573** | 0.464 | **0.433** | **0.865** | 0.847 | 0.794 | **0.539** | **0.398** | **0.302** |

| Model | LoVR-TH | PEV-K | LoVR-V | VDC-D | MS-TI | MS-TV | MSRVTT-I2V | LoVR-C2V |
|---|---|---|---|---|---|---|---|---|
| | R@10 | R@1 | R@1 | R@1 | P@1 | P@1 | R@1 | R@1 |
| CLIP4Clip | 0.338 | 0.179 | 0.360 | 0.566 | 0.173 | 0.183 | **0.924** | 0.503 |
| ViCLIP | 0.202 | 0.075 | 0.230 | 0.395 | 0.283 | 0.243 | 0.846 | 0.433 |
| VideoCLIP-XL | 0.439 | 0.229 | 0.380 | 0.820 | 0.230 | 0.223 | 0.861 | 0.403 |
| LanguageBind | 0.425 | 0.303 | 0.540 | 0.679 | 0.228 | 0.233 | 0.827 | 0.463 |
| InternVideo2-1B | 0.298 | 0.026 | 0.280 | 0.485 | 0.265 | 0.230 | 0.794 | 0.368 |
| InternVideo2-6B | 0.302 | 0.086 | 0.330 | 0.516 | 0.235 | 0.205 | 0.868 | 0.452 |
| GME-2B | 0.446 | 0.354 | 0.530 | 0.839 | **0.350** | **0.340** | 0.827 | 0.366 |
| Unite-2B | 0.445 | 0.355 | 0.570 | 0.792 | 0.250 | 0.233 | 0.863 | 0.445 |
| VLM2Vec-V2 | 0.492 | 0.324 | 0.610 | 0.913 | 0.275 | 0.250 | 0.841 | 0.385 |
| BGE-VL | 0.387 | 0.184 | 0.550 | 0.722 | 0.303 | 0.233 | 0.779 | 0.465 |
| UniME-7B | 0.504 | 0.323 | 0.480 | 0.847 | 0.310 | 0.305 | 0.867 | **0.537** |
| B3-7B | 0.462 | 0.387 | 0.590 | 0.853 | 0.275 | 0.265 | 0.884 | 0.471 |
| GME-7B | 0.523 | 0.396 | **0.710** | 0.865 | 0.348 | 0.333 | 0.860 | 0.370 |
| Unite-7B | **0.555** | **0.440** | 0.620 | 0.871 | 0.278 | 0.230 | 0.883 | 0.448 |
| GVE-3B | 0.522 | 0.330 | 0.610 | 0.918 | 0.340 | 0.268 | 0.891 | 0.403 |
| GVE-7B | 0.542 | 0.413 | 0.680 | **0.948** | 0.343 | 0.280 | 0.899 | 0.415 |

**Metrics.** Our primary metric is Recall@1 (R@1), which measures if the most relevant item is the correct one. For more challenging datasets with fuzzy queries (e.g., CMRB and LoVR-TH), we choose to report Recall@10 (R@10). Additionally, we use Precision@1 (P@1) for the MS-TI and MS-TV with multiple positive candidates.

**Evaluation Implementations.** All models are evaluated under the strict, unified protocol defined in Section 3.1 to ensure fair and reproducible comparisons. To isolate core representational power, we standardize the output pipeline: model parameters are loaded in `bf16` precision, all embeddings are $\ell_2$-normalized, and cosine similarity serves as the universal metric, with no model-specific post-processing or re-ranking. Critically, we remove any learnable projection heads (e.g., the MLP head in the InternVideo2 series), using the hidden state from the final backbone layer as the true general-purpose embedding. On the input side, the evaluation is vision-only, with each video uniformly sampled into exactly 8 frames. To balance fairness with architectural diversity, CLIP-based models process frames at a fixed $224 \times 224$ resolution, while MLLM-based models are constrained to a budget of <200 visual tokens per frame via adaptive resizing. All input sequences are capped at 8192 tokens, and for models without native video support (e.g., BGE-VL), we implement a multi-image pipeline to ensure comparability.

**Others.** We provide complete details in the appendix, covering: (1) the construction and evaluation of UVRB (Appendix A.2-A.3); (2) the data synthesis pipeline with prompts (Appendix A.4-A.6);

Table 2: Video retrieval performance by specific abilities (tasks and domains) on UVRB. The AVG values are averaged over tasks (textual (TXT), composed (CMP), visual (VIS)) and domains (coarse-grained (CG), fine-grained (FG), long-context (LC)) video retrieval tasks. Besides, we provide sub-domain results, including spatial (S), temporal (T), partially relevant (PR). For each column: highest score is **bolded**, second-highest is underlined.

| Model | AVG | Tasks | | | Domains | | | Sub-domains | | |
|---|---|---|---|---|---|---|---|---|---|---|
| | | TXT | CMP | VIS | CG | FG | LC | S | T | PR |
| CLIP4Clip | 0.416 | 0.401 | 0.178 | **0.714** | 0.380 | 0.360 | 0.463 | 0.559 | 0.285 | 0.236 |
| ViCLIP | 0.375 | 0.336 | 0.263 | 0.640 | 0.380 | 0.315 | 0.313 | 0.484 | 0.289 | 0.171 |
| VideoCLIP-XL | 0.510 | 0.550 | 0.227 | 0.632 | 0.558 | 0.493 | 0.600 | 0.787 | 0.381 | 0.310 |
| LanguageBind | 0.508 | 0.543 | 0.231 | 0.645 | 0.539 | 0.479 | 0.610 | 0.723 | 0.378 | 0.336 |
| InternVideo2-1B | 0.420 | 0.422 | 0.248 | 0.581 | 0.480 | 0.403 | 0.383 | 0.606 | 0.413 | 0.189 |
| InternVideo2-6B | 0.445 | 0.448 | 0.220 | 0.660 | 0.504 | 0.417 | 0.423 | 0.631 | 0.400 | 0.220 |
| GME-2B | 0.416 | 0.539 | **0.345** | 0.597 | 0.461 | 0.471 | 0.685 | 0.716 | 0.349 | 0.347 |
| Unite-2B | 0.507 | 0.536 | 0.242 | 0.654 | 0.455 | 0.471 | 0.681 | 0.725 | 0.347 | 0.341 |
| VLM2Vec-V2 | 0.538 | 0.587 | 0.263 | 0.613 | 0.498 | 0.502 | 0.762 | 0.809 | 0.348 | 0.348 |
| BGE-VL | 0.480 | 0.497 | 0.268 | 0.622 | 0.448 | 0.406 | 0.636 | 0.664 | 0.292 | 0.261 |
| UniME-7B | 0.542 | 0.561 | 0.308 | 0.702 | 0.500 | 0.518 | 0.664 | 0.785 | 0.396 | 0.373 |
| B3-7B | 0.538 | 0.570 | 0.270 | 0.678 | 0.482 | 0.505 | 0.722 | 0.797 | 0.364 | 0.355 |
| GME-7B | 0.562 | 0.604 | 0.341 | 0.615 | 0.518 | 0.507 | 0.788 | 0.749 | 0.373 | 0.398 |
| Unite-7B | 0.559 | 0.609 | 0.254 | 0.666 | 0.541 | 0.539 | 0.746 | 0.779 | 0.412 | **0.425** |
| GVE-3B | 0.571 | 0.619 | 0.304 | 0.647 | 0.552 | 0.541 | 0.764 | 0.816 | 0.430 | 0.377 |
| GVE-7B | **0.600** | **0.657** | 0.312 | 0.657 | **0.587** | **0.570** | **0.814** | **0.821** | **0.469** | 0.419 |

Table 3: Ablation study for UVRD and MODALITY PYRAMID. **AVG of D** : average across datasets, **AVG of A** : average across abilities. For each column of each size of model (3B or 7B): highest score is **bolded**, second-highest is underlined.

| Model | D | A | Tasks | | | Domains | | | Sub-domains | | |
|---|---|---|---|---|---|---|---|---|---|---|---|
| | AVG | AVG | TXT | CMP | VIS | CG | FG | LC | S | T | PR |
| GVE-i-3B | 0.528 | 0.558 | **0.620** | 0.237 | 0.632 | 0.532 | 0.521 | **0.808** | 0.781 | 0.402 | **0.379** |
| GVE-s-3B | 0.537 | 0.564 | 0.617 | 0.301 | 0.617 | 0.539 | 0.536 | 0.775 | 0.811 | 0.421 | 0.377 |
| GVE-3B | **0.544** | **0.571** | 0.619 | **0.304** | **0.647** | **0.552** | **0.541** | 0.764 | **0.816** | **0.430** | 0.377 |
| GVE-i-7B | 0.563 | 0.587 | 0.643 | 0.274 | **0.678** | 0.567 | 0.566 | 0.795 | 0.812 | 0.459 | **0.426** |
| GVE-s-7B | 0.568 | 0.594 | 0.648 | **0.313** | 0.662 | 0.576 | 0.563 | 0.804 | 0.814 | 0.458 | 0.418 |
| GVE-7B | **0.573** | **0.600** | **0.657** | 0.312 | 0.657 | **0.587** | **0.570** | **0.814** | **0.821** | **0.469** | 0.419 |

and (3) the GVE model's architecture and training specifics (Appendix A.7-A.8). The appendix also includes baseline properties and more experimental results (Appendix A.9-A.15).

## 4.2 MAIN RESULTS

**Overall Performance.** We evaluate GVE on the UVRB benchmark under a strictly zero-shot setting without any exposure to in-domain data during training. Although competing models may have an unfair advantage for using training data corresponding to several test sets, our results in Table 1 and Table 2 show clear superiority and confirm the strong generalization of GVE. Specifically, GVE-7B achieves state-of-the-art results with mean scores of 0.573 across datasets and 0.600 across task categories. It outperforms Unite-7B by +6.5% and +7.3% even though Unite-7B may have seen in-domain data. GVE-7B leads in every major dimension including TXT at 0.657 versus 0.609, CMP at 0.312 versus 0.254, CG at 0.587, FG at 0.570, and LC at 0.814 versus 0.746. It also leads in fine-grained subdomains with S at 0.821 versus 0.779 and T at 0.469 versus 0.412. Unite-7B shows strength in VIS and PR tasks but underperforms in compositional and temporal reasoning. Its performance is uneven and relies heavily on specific training data. The compact GVE-3B with 3.8B parameters scores 0.571, higher than Unite-7B at 0.559 with over 7B parameters. This shows our gains come from better data synthesis and curriculum design not from model size or data leakage. Our smaller models match or beat larger ones under fair zero-shot evaluation. This advantage translates to top results while Unite-7B's strength on a few datasets reflects narrow capability. Furthermore, we evaluated GVE on video classification benchmarks in Appendix A.15, where our model demonstrates strong performance, confirming the embedding transferability.

**Ablation Studies**    To dissect the individual contributions of UVRD and MODALITY PYRAMID, we conduct a series of ablation studies. First, we analyze their impact in Table 3. We define GVE-i as a baseline trained on public data with random task scheduling, and GVE-s as the same model enhanced with our synthesized UVRD. As the results show, integrating UVRD (GVE-s vs. GVE-i) yields a significant performance gain (+2.3% for GVE-3B). The benefit is particularly pronounced on complex tasks underrepresented in public data, with a remarkable +27% relative improvement on CMP. Subsequently applying the MODALITY PYRAMID curriculum (GVE vs. GVE-s) further enhances performance across the board, confirming the benefit of structured learning.

Second, to specifically validate the hierarchical design of the curriculum, we compare our full model (GVE) against several variants with different training strategies according to their average scores across datasets, presented in Table 4. These variants include random scheduling (GVE-s), a reverse top-down curriculum (GVE-r), and models trained on only video (GVE-ov) or non-video (GVE-nv) data. The performance ranking is that GVE > GVE-s > GVE-ov > GVE-r > GVE-nv. The order offers two key insights. (1) *The learning sequence is critical:* Our structured curriculum (GVE) significantly outperforms both

Table 4: Ablation on curriculum strategies at 3B scale.

| Model Variant | Score |
|---|---|
| GVE (Ours) | **0.571** |
| GVE-s (Random) | 0.564 |
| GVE-r (Reverse) | 0.545 |
| GVE-ov (Video-Only) | 0.550 |
| GVE-nv (Non-Video) | 0.521 |

random (GVE-s) and reversed (GVE-r) scheduling. (2) *A synergy exists between modalities:* The full model (GVE) also surpasses training on uni-modal data sources alone (GVE-ov and GVE-nv). These findings confirm that a principled, hierarchical integration of diverse data modalities is essential toward optimal performance.

**Data Scaling.**    Figure 7 shows the training-time scaling behavior that performance logarithmically improves with more data, but with diminishing returns. We quantify scaling efficiency by fitting a logarithmic model $y = a \ln x + b$ ($x$: data size, $y$: performance) and report the absolute and relative gain per $10\times$ data increase. On average across datasets (abilities), GVE-3B improves by +7.4% (+7.1%) per decade, while GVE-7B gains +5.4% (+5.4%). While GVE-3B exhibits higher scaling efficiency

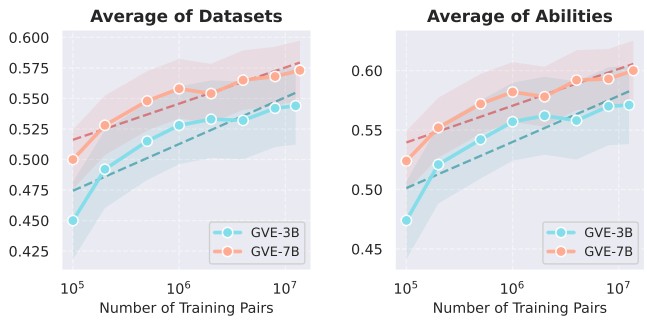

Figure 7: Performance effect from data scaling for GVE series. See Appendix A.11 for detailed results.

in relative terms, GVE-7B starts from a higher baseline, suggesting a trade-off between scaling slope and absolute capability. This result highlights the potential for scaling a more powerful video embedding model with training data at a larger scale. In addition, we also explore the test-time scaling of spatial and temporal density (the frame number and resolution) in Appendix A.12 as a dimension of generalization.

## 4.3    ANALYSIS OF DIMENSIONAL CAPABILITIES

Figures 8 and 9 reveal patterns in how video retrieval models develop capabilities and what their relationships are, as measured primarily by Pearson correlation ($\rho$) for their performance results. Here, we present several discoveries from the analysis in underexplored perspectives.

**Finding 1: Partially Relevant Video Retrieval Better Reflects Universality than Traditional Benchmarks.**    Standard benchmarks such as MSRVTT show a low correlation with average UVR performance ($\rho_{\text{avg}} = 0.58$), suggesting limited representativeness, likely due to overfitting or simplified task design. In contrast, fine-grained (FG), partially relevant (PR), and long-context (LC) retrieval exhibit strong mutual correlations ($\rho \geq 0.90$). More importantly, PR retrieval, although understudied, achieves the highest average correlation with overall performance ($\rho_{\text{avg}} = 0.97$), positioning it as the most reliable proxy for universal capability. We hypothesize this is because successful PR retrieval demands an embedding space that preserves a rich spectrum of information, e.g., from global context

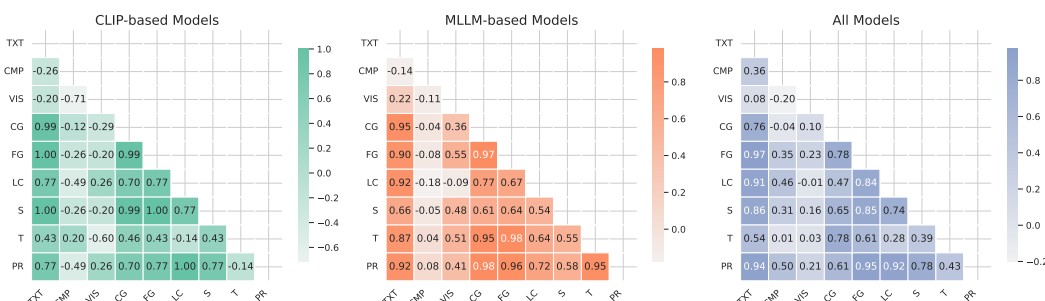

Figure 8: Correlation between dimensional abilities on UVRB for CLIP or MLLM-based models.

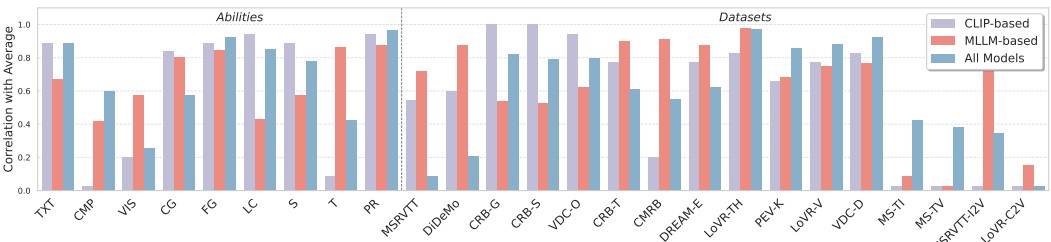

Figure 9: Correlation between averaged performance and abilities or datasets on UVRB.

to subtle, localized details, rather than collapsing video semantics into a single "gist". The ability to maintain this multi-granular semantic richness is a core tenet of universal capability.

**Finding 2: Disentangled Spatial and Temporal Representations.** Models exhibit a marked decoupling between spatial (S) and temporal (T) representation ($\rho = 0.12$). This asymmetry is critical that temporal skills dominate fine-grained understanding ($\rho_{\text{T-FG}} = 0.98$), while spatial skills contribute minimally ($\rho_{\text{S-FG}} = 0.39$). This suggests current works fail to jointly model *when* and *where*, highlighting the need for inductive biases that encourage spatiotemporal integration.

**Finding 3: Performance Divergence Between CLIP and MLLM-based Models.** Model failures are architecture-dependent. For example, CLIP-based models are spatially biased ($\rho_{\text{S-CG}} = 0.99$) but temporally weak ($\rho_{\text{T-CG}} = 0.46$), leading to ability trade-offs. Compositional representation inversely correlates with visual accuracy ($\rho_{\text{CMP-VIS}} = -0.71$). And a near-zero link between temporal and long-context skills ($\rho_{\text{T-LC}} = -0.14$). In contrast, MLLM-based models demonstrate more balanced and integrated learning: superior semantic matching (PR-CG: MLLM $\rho = 0.98$ vs. CLIP 0.70) and stronger temporal-long-context coupling ($\rho_{\text{LC-T}} = 0.64$). Therefore, architecture fundamentally shapes the development of capability, and MLLM has been increasingly popular because of its generalizability in video embedding modeling.

**Finding 4: Scaling Has Limited Impact on Visual Perception.** Parameter scaling improves high-level semantic coherence, but yields negligible gains in low-level visual perception. Notably, the 87M-parameter `CLIP4Clip` (VIS: 0.714) outperforms the 8B-parameter `Unite-7B` (VIS: 0.702). Given the weak correlation between visual fidelity and overall retrieval success ($\rho_{\text{AVG-VIS}} = 0.26$), future progress requires targeted improvements in visual grounding.

## 5 CONCLUSIONS

This work pioneers a unified paradigm for video retrieval. We introduced the first benchmark to comprehensively evaluate dimensional video retrieval abilities. It provides diagnostics to guide us to generate 1.55 million high-fidelity, multi-task training pairs to meet real-world complexity. In addition, we propose a novel curriculum learning algorithm to take advantage of the inherent task-wise relational structure. Based on these, we train a superior MLLM-based video embedding model, GVE. Our experiments validate the state-of-the-art generalization of GVE on UVRB and provide insightful findings in this field. Overall, this paper provides a foundational framework with evaluation-data-training co-design toward a more robust, versatile, and generalizable video retriever.

ETHICS AND REPRODUCIBILITY STATEMENT

This study does not raise concerns related to discrimination, bias, or fairness. To ensure reproducibility, we provide detailed descriptions of the experimental setup in Section 4.1. We present details of evaluations on our benchmark in Appendix A.2, A.3. All data used in our experiments (Appendix A.4) are obtained from previously released and widely adopted datasets, and the data construction and synthesis are also detailed in Appendix A.5, A.6. The model architecture and baseline properties are also specified in Appendix A.7, A.9. In addition, the training implementation and hyperparameters are in Appendix A.8. The presentation of these details can ensure the reproducibility of all results in this work.

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

## A APPENDIX

CONTENT

### A.1 OVERVIEW OF APPENDIX

This appendix provides supplementary material supporting the main paper, organized as follows:

**Methodology Supplementary.**

- **Details of Benchmarking (for Section 3.1):** (1) *UVRB Details (Appendix A.2):* Provides statistics, construction strategies, and evaluation details for all 16 datasets in UVRB. (2) *Evaluation Pipeline (Appendix A.3):* Offers a technical overview of the scalable and reproducible evaluation protocol.

- **Details of Data Synthesis and Preparation (for Section 3.2):** (1) *Training Data (Appendix A.4):* Details the composition and scale of the multi-modal training mixture. (2) *Dataset Construction Pipeline (Appendix A.5):* Explains the design principles of the unified dataset framework. (3) *Synthetic Data Prompts (Appendix A.6):* Presents the structured prompts used for generating synthetic video retrieval data.

- **Details of Model and Training (for Section 3.3):** (1) *Model Architecture (Appendix A.7):* Gives a detailed description of the GVE model architecture, including input fusion and embedding extraction. (2) *Training Implementation (Appendix A.8):* Outlines the parameter-efficient fine-tuning strategy and key hyperparameters.

**Experimental Details and Results.**

- **Baseline Details (Appendix A.9):** Lists the specifications of all 14 evaluated baseline models.
- **Training Dynamics (Appendix A.10):** Visualizes training dynamics and metrics for the GVE-3B and GVE-7B models.
- **Additional Experiments**:
  (1) **Training-time Scaling (Appendix A.11):** Presents the impact of scaling training data on performance, complementing the analysis in the main paper.
  (2) **Test-time Scaling (Appendix A.12):** Analyzes the effect of scaling test-time parameters (e.g., number of frames, resolution) on model performance.
  (3) **Data Quality Validation (Appendix A.13):** Details the human-in-the-loop evaluation conducted to validate the quality of our synthesized UVRD dataset.
  (4) **Curriculum Analysis (Appendix A.14):** Provides in-depth experimental analysis to validate the design and effectiveness of the MODALITY PYRAMID.
  (5) **Video Classification (Appendix A.15):** Reports zero-shot classification performance on standard video action recognition benchmarks.

**Others.**

- **Limitations (Appendix A.16):** Discusses the scope and limitations of the current work and suggests directions for future research.
- **LLM Usage (Appendix A.17):** Discloses the use of Large Language Models as assistive tools during the preparation of this work.

## A.2 UVRB DETAILS

This section provides details of UVRB, including statistics and construction strategies of datasets.

First, the statistics for the number of queries and corpus, the video durations, and the text lengths are presented in Table 5.

Second, we introduce the construction strategies of each dataset as follows.

- **MSRVTT:** We follow the data partition from JSFusion (Yu et al., 2018), utilizing 1,000 clip-text pairs from the original MSRVTT dataset (Xu et al., 2016) for evaluation.
- **DiDeMo:** Following the methodology in (Liu et al., 2019), we concatenate all sentence descriptions associated with a single video from the DiDeMo dataset (Anne Hendricks et al., 2017) to create a paragraph-level query for paragraph-to-video retrieval.
- **CRB-G:** We adhere to the CaReBench protocol (Xu et al., 2025) and use the content from the `caption` field as the general query to retrieve videos.
- **CRB-S:** Similar to CRB-G, we follow (Xu et al., 2025) and select the text from the `spatial_caption` field to form queries focused on spatial descriptions.
- **VDC-O:** We utilize the VDC dataset (Chai et al., 2024) and extract text from the `main_object_caption` field as an object-centric query (e.g., *The main subject, a worker dressed in a gray sleeveless shirt and beige pants...*).
- **CRB-T:** Similar to CRB-S, we again follow (Xu et al., 2025), using the `temporal_caption` field to create queries based on temporal progression.
- **CMRB:** We use the detailed camera motion annotations from CameraBench (Lin et al., 2025) as queries to retrieve videos (e.g., *The camera smoothly dollies forward, maintaining a steady and fluid motion...*).
- **DREAM-E:** For event-based retrieval, we collect event descriptions from the DREAM-1K dataset (Wang et al., 2024b) to serve as queries for event-to-video matching (e.g., *Wooden trap launches purple squirrel into the air*).

Table 5: Statistics of datasets in the Universal Video Retrieval Benchmark (UVRB). All videos use 8 uniformly sampled frames. # Query: the number of queries; # Corpus: the number of corpus; Dur (s): Duration in seconds; # Word: text length in words.

| Dataset | # Query | # Corpus | Dur (s) | # Word |
|---|---|---|---|---|
| **Textual Video Retrieval (Coarse-grained)** | | | | |
| MSRVTT (Xu et al., 2016) | 1,000 | 1,000 | 15.0 | 9.4 |
| DiDeMo (Anne Hendricks et al., 2017) | 1,004 | 1,004 | 53.9 | 29.1 |
| CaReBench-General (CRB-G) (Xu et al., 2025) | 1,000 | 1,000 | 14.4 | 232.2 |
| **Textual Video Retrieval (Fine-grained)** | | | | |
| *(a) Spatial* | | | | |
| CaReBench-Spatial (CRB-S) (Xu et al., 2025) | 1,000 | 1,000 | 14.4 | 115.0 |
| VDC-Object (VDC-O) (Chai et al., 2024) | 1,027 | 1,027 | 30.1 | 91.4 |
| *(b) Temporal* | | | | |
| CaReBench-Temporal (CRB-T) (Xu et al., 2025) | 1,000 | 1,000 | 14.4 | 103.2 |
| CameraBench (CMRB) (Lin et al., 2025) | 728 | 1,071 | 5.7 | 24.8 |
| *(c) Partially Relevant* | | | | |
| DREAM-1K-Event (DREAM-E) (Wang et al., 2024b) | 6,251 | 1,000 | 8.8 | 6.5 |
| LoVR-Theme2Clip (LoVR-TH) (Cai et al., 2025) | 8,854 | 8,854 | 16.9 | 48.1 |
| PE-Video-Keyword (PEV-K) (Bolya et al., 2025) | 14,427 | 15,000 | 16.9 | 45.5 |
| **Textual Video Retrieval (Long-context)** | | | | |
| LoVR-Text2Video (LoVR-V) (Cai et al., 2025) | 100 | 467 | 1560.3 | 17364.5 |
| VDC-Detail (VDC-D) (Chai et al., 2024) | 1,000 | 1,027 | 30.1 | 508.0 |
| **Composed Video Retrieval** | | | | |
| MomentSeeker-Text-Image (MS-TI) (Yuan et al., 2025) | 400 | 10 | 13.5 | 68.5 |
| MomentSeeker-Text-Video (MS-TV) (Yuan et al., 2025) | 400 | 10 | 13.5 | 68.5 |
| **Visual Video Retrieval** | | | | |
| MSRVTT-ImageVideo (MSRVTT-I2V) (Xu et al., 2016) | 1,000 | 1,000 | 15.0 | - |
| LoVR-Clip-to-Video (LoVR-C2V) (Cai et al., 2025) | 467 | 467 | 1560.3 | - |

- **LoVR-TH:** From the LoVR dataset (Cai et al., 2025), we select the theme annotations of video clips as queries for theme-to-clip retrieval (e.g., *The overall style of the animation is vibrant and whimsical...*).

- **PEV-K:** We use the annotations from the `keyword` field in the PE-Video test set (Bolya et al., 2025) to perform keyword-based partially relevant matching (e.g., *colorful, paper, beautiful...*).

- **LoVR-V:** We leverage the full-length video captions from the LoVR dataset (Cai et al., 2025) to perform long-text to long-video retrieval.

- **VDC-D:** For this task, we extract the long, detailed descriptions from the `detailed_caption` field of the VDC dataset (Chai et al., 2024) to serve as fine-grained queries.

- **MS-TI:** Following the adaptation method in (Meng et al., 2025) for the MomentSeeker dataset (Yuan et al., 2025), we create a text-image composed retrieval task. The goal is to retrieve a target video clip using a combined text and image query.

- **MS-TV:** Similar to MS-TI, the task for MS-TV is to use a composed query of a text description and a reference video clip to retrieve the target clip.

- **MSRVTT-I2V:** We construct image-to-video retrieval pairs from the aforementioned MSRVTT test set. For each video, a single frame is randomly sampled to serve as the image query for retrieving its source video.

- **LoVR-C2V:** We leverage the original clip-to-video structure of the LoVR dataset (Cai et al., 2025). For each full-length video, one of its corresponding short clips is used as the query for clip-to-video retrieval.

Third, we list the query prompt and metrics for datasets in evaluations in Table 6.

Table 6: Query Prompts and Metrics for Datasets in UVRB.

| Dataset | Query Prompt | Metric |
|---|---|---|
| MSR-VTT | Find the clip that corresponds to the described scene in the given video. | Recall@1 |
| DiDeMo | Find a video that includes the following described scenes. | Recall@1 |
| CRB-G | Find the video according to the general text description. | Recall@1 |
| CRB-S | Find the video according to the spatial description. | Recall@1 |
| VDC-O | Find the video according to the object description. | Recall@1 |
| CRB-T | Find the video according to the temporal description. | Recall@1 |
| CMRB | Find the video according to the camera motion description. | Recall@10 |
| DREAM-E | Find the video according to the text description. | Recall@1 |
| LoVR-TH | Find the video according to text description about video theme information. | Recall@10 |
| PEV-K | Find the video according to the text description of a series of keywords. | Recall@1 |
| LoVR-V | Find the long video according to the long text description. | Recall@1 |
| VDC-D | Find the video according to the detailed text description. | Recall@1 |
| MS-TI | Find the video clip that corresponds to the given text and the given image. | Precision@1 |
| MS-TV | Find the video clip that corresponds to the given text and the given video. | Precision@1 |
| MSRVTT-I2V | Find the video according to the image. | Recall@1 |
| LoVR-C2V | Find the original long video according to the short video clip. | Recall@1 |

## A.3 EVALUATION PIPELINE.

Our evaluation is built atop MTEB (Muennighoff et al., 2022). It decouples the evaluation engine from model architecture—supporting everything from sentence transformers to custom multimodal encoders—via a standardized interface. The pipeline operates in three phases: (1) dynamic task orchestration, (2) configurable model execution, and (3) generation of metrics and diagnostics. For instruction-sensitive tasks, it dynamically injects domain-specific prompts to mirror real-world conditions for actionable diagnostics. Custom benchmarks are integrated seamlessly through dedicated loaders. In this framework, expensive tasks are deferred for efficiency, model initialization (precision, multi-GPU) is abstracted via a factory, and data loading supports both online and offline modes for robustness across environments. Engineering safeguards, including explicit multi-GPU management, clean shutdowns, and offline, first retry logic—ensure reliability at scale.

## A.4 TRAINING DATA

To build a robust and versatile multimodal retrieval system, our models are trained on a large-scale, diverse mixture of text, image, and video data. All tasks are uniformly formulated as instruction-guided retrieval, a strategy designed to foster a unified representation space that can adeptly handle a wide array of queries. The training data is organized into two primary collections: a main mixture of widely-used public datasets (Table 7), and a synthesized set from our Universal Video Retrieval Dataset (UVRD) suite (Table 8).

We follow existing works (Li et al., 2023; Zhang et al., 2025) to construct text-only (e.g., MSMARCO) and image-centric data (e.g., CIRR). In addition, we prepare video datasets using the following strategies.

- **VAST** (Chen et al., 2023): We randomly select one sentence from the multiple vision captions as the text query for each video.
- **InternVid-FLT** (Wang et al., 2023): We drop about 300K low-quality videos and use the left text-video pairs.
- **PE-Video** (Bolya et al., 2025): We choose the human_caption refined on model_caption as the textual description of videos.
- **WebVid** (Bain et al., 2021): We refuse the queries captioning over one video to impede the generality of text, which excludes 5M videos in final training approximately.

As the aforementioned methodology, we train our model based on contrastive learning with pre-mined explicit hard negatives and in-batch negatives. To maintain a balanced training diet and manage

Table 7: Configuration of the main training data mixture for our 3B and 7B models. 'K' denotes thousands, 'M' denotes millions.

| Dataset | Task[a] | Sample Size | | Neg.[b] | BS |
|---|---|---|---|---|---|
| | | 3B | 7B | | |
| **Part 1: Text-only Data** | | | | | |
| MSMARCO | T→T | 300K | 500K | 2 | 64 |
| HotpotQA | T→T | 69K | | 1 | 64 |
| WebQA | T→T | 11K | | 1 | 32 |
| **Part 2: Image-centric Data** | | | | | |
| CIRR | TI→T | 16K | | 1 | 32 |
| Fashion200K | T→I | 4K | | 1 | 32 |
| Nights | I→I | 13K | | 1 | 32 |
| OVEN | TI→TI | 20K | 30K | 1 | 64 |
| OVEN | TI→T | 20K | 30K | 1 | 64 |
| VisualNews | T→I | 40K | 60K | 1 | 64 |
| EDIS | T→TI | 12K | | 1 | 32 |
| FashionIQ | TI→I | 4K | | 1 | 32 |
| MSCOCO | I→T | 4K | | 1 | 32 |
| REMUQ | TI→T | 5K | | 1 | 32 |
| WebQA | T→TI | 12K | | 1 | 32 |
| LLAVA | TI→T | 20K | 30K | 1 | 64 |
| EVQA | TI→TI | 20K | 30K | 1 | 64 |
| CC3M | T→I | 200K | 300K | 3 | 32 |
| CC3M | I→T | 100K | 200K | 2 | 64 |
| Laion | T→I | 300K | 500K | 3 | 32 |
| Laion | I→T | 200K | | 2 | 64 |
| ImageNet | I→T | 100K | 200K | 2 | 32 |
| VL3-Syn7M (short) | T→I | 300K | 500K | 3 | 32 |
| VL3-Syn7M (short) | I→T | 100K | 200K | 2 | 64 |
| VL3-Syn7M (detailed) | T→I | 200K | 300K | 3 | 32 |
| VL3-Syn7M (detailed) | I→T | 100K | 200K | 2 | 64 |
| VISTA | TI→I | 100K | | 2 | 32 |
| VISTA | T→TI | 20K | | 1 | 64 |
| **Part 3: Video-centric Data** | | | | | |
| VAST | T→V | 1.6M | | 0 | 32 |
| InternVid-FLT | T→V | 1.7M | | 0 | 32 |
| PE-Video | T→V | 104K | | 1 | 64 |
| WebVid | T→V | 5.4M | | 0 | 32 |

[a] **Task**: T=Text, I=Image, V=Video. The format is Query→Corpus.
[b] **Neg.**: Number of explicit hard negatives per positive. '0' indicates use of in-batch negatives only.

Table 8: Configuration of UVRD. Sample sizes are identical for both 3B and 7B models.

| Dataset | Task[a] | Sample Size | Neg.[b] | BS |
|---|---|---|---|---|
| UVRD-T2T | T→T | 100K | 0 | 64 |
| UVRD-T2I | T→I | 210K | 0 | 32 |
| UVRD-T2V | T→V | 879K | 1 | 64 |
| UVRD-TI2V | TI→V | 89K | 1 | 64 |
| UVRD-TV2V | TV→V | 35K | 1 | 64 |
| UVRD-I2V | I→V | 200K | 0 | 64 |
| UVRD-V2V | V→V | 36K | 0 | 64 |

[a] **Task**: T=Text, I=Image, V=Video. The format is Query→Corpus.
[b] **Neg.**: Number of explicit hard negatives per positive. '0' indicates use of in-batch negatives only.

computational load, we employ a sophisticated data sampling strategy. For extremely large datasets, we sample a fixed number of instances. To amplify the learning signal from certain high-value or complex datasets, we may apply an upsampling strategy by repeating their data.

The scale of our training data is substantial. For our 3B parameter model, we prepared 12.55 million instances. To further leverage the capacity of our 7B parameter model, we increase the sampling rate for several large-scale datasets, bringing the total instances to 13.73 million. A detailed breakdown of the data composition is provided in Table 9.

Table 9: Total number of instances for our 3B and 7B models across all data categories.

| Data Category | Total Sample Size | |
| --- | --- | --- |
| | **3B Model** | **7B Model** |
| Collected: Text-only | 380K | 580K |
| Collected: Image-centric | 1.82M | 2.80M |
| Collected: Video-centric | 8.80M | 8.80M |
| Synthesized: UVRD (All Modalities) | 1.55M | 1.55M |
| **Total** | **12.55M** | **13.73M** |

## A.5 DATASET CONSTRUCTION PIPELINE DETAILS

Current multimodal retrieval research is hindered by fragmented, ad-hoc dataset construction. We introduce a unified, object-oriented framework that elevates this process to a rigorous science, balancing conceptual clarity with engineering robustness for scalable, reproducible benchmarking.

The framework is anchored in a canonical tripartite abstraction: every retrieval task is decomposed into a CORPUS, a set of QUERIES, and a RELEVANCE MAPPING. This schema, enforced by an abstract base class, is a conceptual invariant that ensures structural consistency across all datasets, from text-to-video to complex composed queries, enabling seamless model and evaluation compatibility.

The expressiveness of this data construction pipeline stems from polymorphic specialization via inheritance, encapsulated in three core strategies:

**Modality as Configuration.** Modality is a dynamic parameter. By overriding a single method, a dataset effortlessly transitions between modalities (text, image, video, or composite), transforming benchmarks like MSRVTT into image-to-video tasks with minimal code.

**Task Derivation via Inheritance.** Complex tasks are composed of simpler ones. A text+image-to-video task inherits its base structure and extends the query schema. Variants are derived by overriding specific data-access methods, turning benchmark creation into a modular, hypothesis-driven workflow.

**Engineering for Scale and Integrity.** Scalability and data quality are first-class principles. Large datasets (e.g., InternVid-FLT) leverage chunked, parallel processing. Crucially, proactive curation is embedded: automated validators check video integrity (resolution, frame count), while repair mechanisms fix common errors using `ffmpeg`, ensuring failures reflect retrieval challenges, i.e, not data corruption.

## A.6 PROMPT FOR SYNTHETIC VIDEO RETRIEVAL DATA GENERATION

To ensure high-quality, diverse, and controllable synthetic video retrieval data generation at scale, we design and deploy a suite of structured prompts to instruct MLLMs for captioning. Our pipeline operates in four parts:

1. **Raw (Or Weakly Annotated) Video (Re-) Captioning**: Enhance raw or uncaptioned videos with rich, diverse textual descriptions.

2. **Text-Image Composed Retrieval**: Generate queries that combine reference images with video content for fine-grained retrieval.

3. **Text-Video Composed Retrieval**: Generate queries that combine short reference clips with target videos for practical temporal or perspective-based retrieval.

4. **Frame Image Captioning**: Annotate individual video frames with dynamic-aware captions for auxiliary training signals.

All prompts enforce strict output formatting, factual grounding, and stylistic diversity to ensure dataset quality and coverage. {raw_caption} represents an optional, potentially low-quality human-provided or auto-generated caption associated with the video. Placeholders (e.g., {readability} and {education_level}) are dynamically instantiated during batch generation.

Besides, we can generate text-to-text retrieval data from the multiple video/frame captions by randomly matching any two sentences in the caption list.

---

**⊞ Synthetic Video Captioning Prompt**

```
Generate 5 distinct and high-quality ENGLISH captions for the provided
    video. Please first visually understand the video file and analyze
    the video frame-by-frame in depth before captioning.

The original video caption is {raw_caption}. (Ignore if empty.)

Each caption must be:
  1. The final answer MUST only be a JSON dict of captions where the
    key is the caption number and the value is a single-paragraph
    caption.
  2. Factually accurate and descriptive - include only what is clearly
    visible or reasonably inferable from the video.
  3. Focus exclusively on visible content; do not mention absences or
    speculate about unseen elements.

Content Requirements (per caption):
  1. Spatial Details (30-60%): Describe location, setting, key
    environmental elements, objects, and their spatial relationships.
    Include notable visual features.
  2. Temporal Flow (30-60%): Capture event sequence and action dynamics
     - movements, interactions, transitions - as they unfold over time.
  3. Theme/Background/Style/Meaning/Highlight/Camera (0-20%): Describe
    observable emotional tone, narrative style, thematic elements,
    highlighting frames, or significant camera movements/angles.
  4. Others (0-10%).

Key Guidelines:
  1. Do not invent fictional elements, dialogue, or backstory not
    visible in the video.
  2. Use the following varied sentence styles (one per caption):
        - concise and punchy summary,
        - spatial-temporal richly descriptive,
        - abstract understanding,
        - keywords-only,
        - partially relevant information.
  3. Ensure diversity: avoid repetition in wording, focus, or rhythm;
    at least one caption must be <20 words and one >100 words; balance
    objectivity with vivid sensory language; randomly omit minor
    details in 1 or 2 captions.
  4. Ensure readability matches {readability} and is appropriate for {
    education_level} readers.

Now generate these captions in strict JSON format:
{
   "1": <caption text 1: concise and punchy summary, 10-25 words>,
   "2": <caption text 2: spatial-temporal richly descriptive, 80-200
   words>,
   "3": <caption text 3: abstract understanding, 30-100 words>,
   "4": <caption text 4: keywords-only, 5-30 words>,
   "5": <caption text 5: partially relevant information, 10-70 words>
}
```

**🤖 Synthetic Text-Image Composed Video Retrieval Prompt**

```
You are an information retrieval expert specializing in high-value text
    -image composed queries. Your sole objective is to generate queries
     that significantly enhance video retrieval performance by
    effectively combining a reference image and video content.

## What Makes a HIGH-QUALITY Query (Non-Negotiable)
A truly valuable query must satisfy ALL of these criteria:

1. COMBINATION NECESSITY (Most Critical)
   - The query MUST become meaningless or significantly less specific
    if either the image or video is removed.
   - Example (HIGH-QUALITY): "the person FROM REFERENCE IMAGE wearing
    red jacket now skiing"
   - Example (LOW-QUALITY): "a person skiing" (works without image)

2. SEMANTIC PRECISION
   - Must accurately reflect BOTH the visual content of the reference
    image AND the video.
   - Must reference at least ONE specific visual attribute from the
    image (not generic descriptions).
   - Example (HIGH-QUALITY): "matching the blue hat FROM PHOTO, now
    running through park"
   - Example (LOW-QUALITY): "someone similar to image moving" (too
    vague)

3. RETRIEVAL EFFECTIVENESS
   - Must narrow search results by at least 50\% compared to text-only
    queries.
   - Must contain actionable constraints that differentiate from 90\%
    of videos in the database.
   - Example (HIGH-QUALITY): "the woman WITH PONYTAIL FROM IMAGE
    entering building at 2PM"
   - Example (LOW-QUALITY): "a woman walking" (too broad)

4. LOGICAL COHERENCE
   - Must maintain subject-verb-object consistency across modalities.
   - Must avoid semantic contradictions between image attributes and
    video actions.
   - Example (HIGH-QUALITY): "the dog FROM REFERENCE PHOTO chasing a
    ball"
   - Example (LOW-QUALITY): "the red jacket FROM IMAGE running down
    hill" (jackets don't run)

5. PRACTICAL UTILITY
   - Must solve a real-world ambiguity that neither text nor image
    could resolve alone.
   - Must reflect how actual users would express their information need.

   - Example (HIGH-QUALITY): "same person AS IN PHOTO but wearing blue
    instead of red"
   - Example (LOW-QUALITY): "the image shows a person and the video
    shows action" (no real combination)

## High-Value Query Generation Framework
Follow this structured approach:

1. DEEP ANALYSIS PHASE (Mandatory)
   a) Reference Image Analysis:
      - Identify 1-4 SPECIFIC visual attributes (e.g., "red jacket", "
    ponytail", "blue hat")
      - Determine primary subject with discriminative features
      - Note what CANNOT be determined from image (e.g., action, scene)

   b) Video Content Analysis:
```

```
        - Identify primary action using precise verbs (e.g., "running", "
     entering", "chasing")
        - Note scene context and temporal elements
        - Determine what CHANGES from the static image reference

2. COMBINATION STRATEGY SELECTION
   Choose ONE primary strategy:

   A) IDENTITY PRESERVATION + ACTION CHANGE
      - Structure: [Binding phrase] + [Image attribute] + [Video action
     ]
      - Example: "the man FROM REFERENCE IMAGE in blue shirt now
     running"

   B) IDENTITY PRESERVATION + SCENE MIGRATION
      - Structure: [Binding phrase] + [Image attribute] + [Scene
     transition]
      - Example: "same person AS IN PHOTO moving from office to park"

   C) IDENTITY PRESERVATION + NEGATIVE CONSTRAINT
      - Structure: [Binding phrase] + [Negative constraint] + [Video
     state]
      - Example: "not wearing red jacket FROM IMAGE but blue, walking"

   D) RELATIONAL TRANSFER
      - Structure: [Binding phrase] + [Relationship description]
      - Example: "the dog FROM REFERENCE PHOTO chasing a ball"

3. QUALITY ENHANCEMENT TECHNIQUES
   - Binding Precision: Use "FROM REFERENCE IMAGE", not "like the
    picture"
   - Attribute Specificity: Use concrete features ("red jacket", not "
    clothing")
   - Action Verbs: Use present continuous tense ("running", not "runs")
   - Context Enrichment: Add 1 relevant scene descriptor ("in park", "
    near building")
   - Noise Handling: For low-quality inputs, use "resembling" but
    maintain specificity
   - Audience Adaptation: Ensure readability matches {readability} and
    suits {education_level} readers.

4. MANDATORY QUALITY CHECK
   Before finalizing, verify ALL:
   - [ ] Explicit binding to reference image
   - [ ] References SPECIFIC visual attribute from image
   - [ ] Describes DYNAMIC ELEMENT not in static image
   - [ ] Loses specificity if image removed
   - [ ] Contains actionable retrieval constraint
   - [ ] Maintains logical subject-action consistency
   - [ ] Solves real-world ambiguity

## What to AVOID
- Generic descriptions ignoring image specificity
- Redundant mentions of obvious image content
- Semantic contradictions (e.g., "the jacket is running")
- Overly precise details not visible in inputs
- Standalone video descriptions without image binding
- Vague terms: "something", "thing", "area"
- Excessive length without added value (>50 words)

## Output Format
- Output ONLY a JSON object with one key: "query"
- Value must be the generated query sentence
- No additional text or formatting

Generate the query for:
Reference Image: <image_input>
```

```
Video Clip: <video_input>
```

**🤖 Synthetic Text-Video Composed Video Retrieval Prompt**

```
You are an information retrieval expert specializing in practical text-
    video composed queries. Your objective is to generate queries that
    effectively combine a short reference clip with a target video for
    real-world retrieval.

## Practical Context Understanding
In real-world scenarios:
- Reference is typically a SHORT CLIP (2-10 seconds)
- Clip is usually HIGHLY RELEVANT to target video (same source or
    similar content)
- Common relationships: temporal continuation, perspective variation,
    quality differences, minor action variations
- Query should reflect how users actually search

## What Makes a HIGH-QUALITY Query (Practical Focus)
Must satisfy these criteria:

1. USEFUL COMBINATION (Most Important)
    - Leverages reference clip to specify what text alone cannot
    - Example (HIGH-QUALITY): "the same person FROM REFERENCE CLIP
    continuing to run after the jump"
    - Example (LOW-QUALITY): "a person running" (ignores reference)

2. PRACTICAL PRECISION
    - References at least ONE observable feature from reference clip
    - Example (HIGH-QUALITY): "matching the red jacket FROM REFERENCE,
    now entering building"
    - Example (LOW-QUALITY): "someone similar to clip moving" (too vague
    )

3. REAL-WORLD UTILITY
    - Helps find videos difficult to retrieve with text alone
    - Example (HIGH-QUALITY): "same action AS IN REFERENCE but from
    front angle"
    - Example (LOW-QUALITY): "the clip shows action" (no retrieval value
    )

4. NATURAL EXPRESSION
    - Sounds like how a real user would phrase it
    - Example (HIGH-QUALITY): "what happens right after this moment?"
    - Example (LOW-QUALITY): "temporal continuation of the current
    visual sequence" (too academic)

## Practical Query Generation Framework

1. REFERENCE CLIP ANALYSIS
    - Identify 1-3 KEY OBSERVABLE FEATURES (e.g., "red jacket", "
    starting pose", "mid-action")
    - Determine most distinctive visual or action element
    - Note what is CLEARLY VISIBLE (avoid guessing)

2. TARGET VIDEO RELATIONSHIP ASSESSMENT
    Determine relationship type:

    A) TEMPORAL CONTINUATION
        - Reference is earlier part of same sequence
        - Query focus: "what happens next" or "continuing action"

    B) PERSPECTIVE VARIATION
        - Same action from different angle/view
        - Query focus: "same action from different angle"
```

```
    C) QUALITY/CONDITION VARIATION
        - Same action with different lighting/resolution
        - Query focus: "same scene in better lighting"

    D) MINOR ACTION VARIATION
        - Slightly different execution of similar action
        - Query focus: "same person but running instead of walking"

3. QUERY CONSTRUCTION
    - Start with binding phrase: "FROM REFERENCE CLIP", "AS IN REFERENCE
    ", etc.
    - Reference 1-2 specific observable features from clip
    - Describe relationship to target video clearly
    - Keep natural and practical (5-15 words typically)
    - For temporal: focus on "what happens next"
    - For perspective: specify desired viewpoint
    - For quality: specify desired condition
    - For action: specify the change

4. PRACTICAL QUALITY CHECK
    Ask before finalizing:
    - Would this help me find what I'm looking for?
    - Does it add value beyond describing the target?
    - Would a real user phrase it this way?
    - Is everything mentioned clearly visible in reference?

## What to AVOID
- Overly academic or technical language
- References to features NOT clearly visible
- Excessive precision about timing ("exactly 3.2 seconds later")
- Queries that work equally well without reference
- Generic descriptions: "similar video", "related content"
- Making unsupported assumptions

## Output Format
- Output ONLY a JSON object with key: "query"
- Value must be the generated query string
- No additional text, explanations, or formatting

Generate the query for:
Reference Clip: <reference_clip_input>
Target Video: <target_video_input>
```

### 🖼 Synthetic Frame Captioning Prompt

```
Generate 5 distinct and high-quality ENGLISH captions for the provided
    image (a frame extracted from a video) based solely on visual
    content. Please first visually understand the image and analyze it
    in depth before captioning.

Each caption must be:
  1. The final answer MUST only be a JSON dict of captions where the
    key is the caption number and the value is a single-paragraph
    caption.
  2. Factually accurate and descriptive - include only what is clearly
    visible; captions should also be consistent with the short-term
    video context.
  3. Focus exclusively on visible content; do not mention absences or
    speculate about unseen context.

Content Requirements (per caption):
  1. Spatial Details (50-70%): Describe location, setting, key objects,
     spatial relationships, colors, lighting, composition. Include
    instantaneous action states derived from visible posture/motion
    cues.
```

```
   2. Temporal Snapshots (0-30%): Describe the frozen moment's temporal
      state (movements, interactions, transitions) ONLY if visually
      provable. Avoid implying sequence, duration, or speed.
   3. Theme/Style/Composition (10-20%): Cover emotional tone, camera
      angle, lighting style, or artistic elements directly observable.
   4. Others (0-10%).

 Key Guidelines:
   1. Do not invent fictional elements or backstory. If action is
      ambiguous, describe neutrally. Never use future/past tense or
      speculative phrases.
   2. Use varied sentence styles (randomly assign one per caption):
          - concise spatial-temporal snapshot,
          - spatially rich descriptive,
          - abstract spatial interpretation,
          - keywords with temporal anchors,
          - minimalist spatial focus.
   3. Ensure diversity: avoid repetition in wording, focus, or rhythm;
      at least one caption <20 words and one >100 words; balance
      objectivity with vivid sensory language; randomly omit minor
      details in 1-2 captions.
   4. Ensure readability matches {readability} and is appropriate for {
      education_level} readers.

 Now generate these captions in strict JSON format:
 {
    "1": <caption text 1: concise and punchy summary, 10-25 words>,
    "2": <caption text 2: spatial-temporal richly descriptive, 50-200
    words>,
    "3": <caption text 3: abstract understanding, 30-100 words>,
    "4": <caption text 4: keywords-only, 5-30 words>,
    "5": <caption text 5: partially relevant information, 10-70 words>
 }
```

**Prompt Utilization.** These prompts are designed for a scalable, schema-constrained generation pipeline for synthesizing diverse video-centric annotations and queries. The system unifies four tasks under a single modular framework. Each task is governed by a structured prompt template with dynamic control slots (e.g., readability, education level), ensuring linguistic and semantic diversity. Input modalities (image, video, or both) are automatically routed and embedded via a distributed multimodal LLM (Keye-VL-8B (Team et al., 2025), 32K context), with outputs rigorously validated against JSON schemas for structural correctness. The pipeline supports sharded, resumable batch generation with quality-aware sampling, enabling the production of millions of grounded, stylistically varied synthetic instances.

### A.7 MODEL ARCHITECTURE DETAILS

The GVE model is architecturally derived from Qwen2.5-VL (Bai et al., 2025), repurposed as a fixed-length multimodal encoder by removing its autoregressive head. Its core function is to map arbitrarily composed inputs—text, image, or video—into a shared $d$-dimensional embedding space, preserving cross-modal alignment inherited from pretraining.

Input fusion begins with tokenization and visual encoding. Text is converted into a sequence of token IDs $\mathbf{X}_t \in \mathbb{Z}^{B \times T_t}$, while images and video frames are encoded into visual token sequences $\mathbf{X}_v \in \mathbb{R}^{B \times T_v \times d}$, where $T_v = \text{THW}/p_s^2$ for images, and $T_v = K \cdot (\text{THW}/p_s^2)$ for videos with $K$ uniformly sampled frames. Note that frame embeddings will be added with absolute time encoding. Critically, $K$ must satisfy $K \bmod p_t = 0$ (default: $p_t = p_s = 2$) to maintain alignment with the vision encoder's 3D spatiotemporal grid. The processor then injects these visual tokens into the textual sequence by replacing placeholder tokens (`<image>`, `<video>`), producing a fused input $\mathbf{X}_{\text{fused}} \in \mathbb{R}^{B \times T \times d}$, where $T = T_t + \sum T_v$. This scatter-based fusion preserves positional coherence and enables interleaved modality composition, which is essential for compositional query understanding in multimodal embeddings. The final embedding $\mathbf{e}^{(i)} \in \mathbb{R}^d$ for the $i$-th instance is

extracted from the last attended token in the sequence:

$$\mathbf{e}^{(i)} = \frac{\mathbf{h}_{p_i}^{(i)}}{\|\mathbf{h}_{p_i}^{(i)}\|_2}, \quad \text{where } p_i = \max\{j \mid \mathbf{M}_j^{(i)} = 1\},$$

and $\mathbf{M}^{(i)}$ is the attention mask for instance $i$. This position corresponds to the EOS token in left-padded sequences or the final non-pad token in right-padded ones. The choice is motivated by the observation that in instruction-tuned MLLMs, the final token often encapsulates the model's response intent, making it semantically aligned with the user's retrieval goal.

## A.8 TRAINING IMPLEMENTATION AND HYPERPARAMETERS

**Parameter-Efficient Tuning.** To facilitate parameter-efficient fine-tuning (PEFT), we employ Low-Rank Adaptation (LoRA) (Hu et al., 2021). Our strategy involves a targeted application of LoRA to the language components of the model, specifically the q_proj, v_proj, k_proj, up_proj, down_proj, and gate_proj modules. Crucially, the entire visual backbone and the base token embedding layer are kept frozen. This approach focuses adaptation on high-level semantic and cross-modal reasoning while preserving the powerful pretrained visual features. We enable FlashAttention-2 (Dao, 2023) to accelerate training and reduce memory. The final embedding for each input is obtained via last-token pooling on the last hidden layer, followed by L2 normalization to project embeddings onto the unit hypersphere for stable cosine similarity computations.

**Optimizer and Training Dynamics.** The model is trained using the AdamW (Loshchilov & Hutter, 2017) optimizer with a learning rate of $3 \times 10^{-5}$ and a weight decay of $0.1$. A cosine learning rate scheduler is used. Training is performed in BFloat16 (bf16) mixed-precision. The entire training process is managed under the DeepSpeed framework. For video inputs, we uniformly sample 8 frames per clip at a rate of $1.0$ FPS. By default, we use 32 NVIDIA A100 GPUs, each with 80GB of memory. Therefore, the overall batch size is at least 1024.

**Memory Optimization and Distributed Strategy.** To manage GPU memory, we enable gradient checkpointing, which recomputes intermediate activations during the backward pass. While our framework supports more advanced techniques like Gradient Cache, it was disabled in favor of this standard approach. To leverage our multi-GPU setup, we enable cross-device negative sharing. This strategy gathers embeddings from all GPUs, effectively multiplying the pool of in-batch negatives by the number of devices. This enriches the negative set for the contrastive loss computation on each GPU, leading to a stronger training signal without increasing per-device memory load.

**Contrastive Learning and Stability.** The core of our training is an InfoNCE-style contrastive loss with a temperature of $0.03$. The use of cross-device negatives starts from the first step. We also enhance our training logs with contrastive-specific metrics, including the average scores of positive pairs and the average margin between positive and hard-negative pairs, providing crucial insights into the model's learning dynamics.

**Hyperparameter Summary.** A comprehensive summary of all key hyperparameters is provided in Table 10.

## A.9 BASELINE DETAILS

Here we present more details of baseline models tested on our benchmark in Table 11, including full model names, abbreviations, architectures, sizes, and training data types. Based on these properties, we analyze to discover potential performance knowledge and dependencies in our experimental part.

## A.10 TRAINING DYNAMICS

We monitor four key metrics during training: *Training Loss*, *Mean Score*, *Max Negative Gap*, and *Mean Positive Score* in Figure 10 and Figure 11. To ensure robust visualization, we mitigate outlier effects, followed by 200-step moving average smoothing. Original trajectories (subsampled every 100 steps) are plotted alongside smoothed trends, with ±1 standard deviation bands indicating local volatility.

Table 10: Key hyperparameters used for model training.

| Parameter | Value |
|---|---|
| **Model & LoRA Architecture** | |
| Base Model | `Qwen2.5-VL-3B-Instruct` or `Qwen2.5-VL-7B-Instruct` |
| PEFT Method | LoRA |
| LoRA Rank ($r$) | 16 |
| LoRA Alpha ($\alpha$) | 32 |
| LoRA Dropout | 0.1 |
| LoRA Target Modules | `q_proj, v_proj, k_proj, up_proj, down_proj, gate_proj` |
| Frozen Components | Visual Backbone, Token Embeddings |
| **Optimizer & Training** | |
| Optimizer | AdamW |
| Learning Rate | $3 \times 10^{-5}$ |
| Weight Decay | 0.1 |
| LR Scheduler | Cosine |
| Training Epochs | 3 |
| Precision | BF16 |
| Gradient Checkpointing | Enabled |
| Seed | 42 |
| **Contrastive Learning** | |
| Temperature | 0.03 |
| In-batch Negatives | Enabled |
| Cross-Device Negatives | Enabled |
| **Data & Preprocessing** | |
| Video Frames per Clip | 8 |
| Video Sampling Rate | 1.0 FPS |
| Dataloader Workers | 1 |

Table 11: Model Abbreviations, Architectures, Parameter Sizes, and Training Data Types. The checkmark (✓) indicates the model was trained on the corresponding data pair types.

| | | | | Contrastive Training Data Pairs | | |
|---|---|---|---|---|---|---|
| **Full Model Name** | **Abbreviation** | **Architecture** | **Size** | **Text-Text** | **Text-Image** | **Text-Video** |
| CLIP4Clip | CLIP4Clip | CLIP-based | 87M | - | - | ✓ |
| ViCLIP | ViCLIP | CLIP-based | 0.4B | - | - | ✓ |
| VideoCLIP-XL | VideoCLIP-XL | CLIP-based | 0.4B | - | - | ✓ |
| LanguageBind-Video-Huge-V1.5 | LanguageBind | CLIP-based | 1.2B | - | - | ✓ |
| InternVideo2-Stage2-1B | InternVideo2-1B | CLIP-based | 1.4B | - | - | ✓ |
| InternVideo2-Stage2-6B | InternVideo2-6B | CLIP-based | 6.4B | - | - | ✓ |
| gme-Qwen2-VL-2B-Instruct | GME-2B | MLLM-based | 2.2B | ✓ | ✓ | - |
| Unite-Base-Qwen2-VL-2B | Unite-2B | MLLM-based | 2.2B | ✓ | ✓ | ✓ |
| VLM2Vec-V2.0 | VLM2Vec-V2 | MLLM-based | 2.2B | ✓ | ✓ | ✓ |
| BGE-VL-v1.5-mmeb | BGE-VL | MLLM-based | 7.6B | ✓ | ✓ | - |
| UniME-LLaVA-OneVision-7B | UniME-7B | MLLM-based | 8.0B | ✓ | ✓ | - |
| B3-Qwen2-7B | B3-7B | MLLM-based | 8.3B | ✓ | ✓ | - |
| gme-Qwen2-VL-7B-Instruct | GME-7B | MLLM-based | 8.3B | ✓ | ✓ | - |
| Unite-Base-Qwen2-VL-7B | Unite-7B | MLLM-based | 8.3B | ✓ | ✓ | ✓ |
| GVE-3B | GVE-3B | MLLM-based | 3.8B | ✓ | ✓ | ✓ |
| GVE-7B | GVE-7B | MLLM-based | 8.3B | ✓ | ✓ | ✓ |

## A.11 EXPERIMENTS OF TRAINING-TIME SCALING: MORE RESULTS OF DATA SCALING

Along with Section 4.2, we depict more experiments for six abilities in data scaling for `GVE-3B` and `GVE-7B` in Figure 12. Shaded bands show $\pm 1$ std; dashed curves are log-linear fits for visual guidance. X-axis is log-scaled to reflect scaling law dynamics.

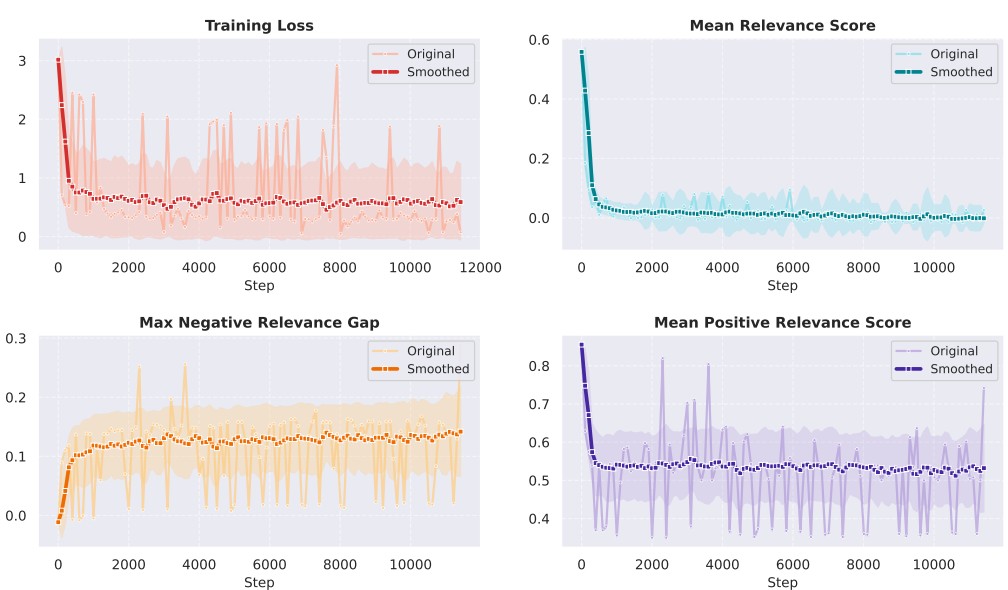

Figure 10: Training dynamics of `GVE-3B` across four metrics.

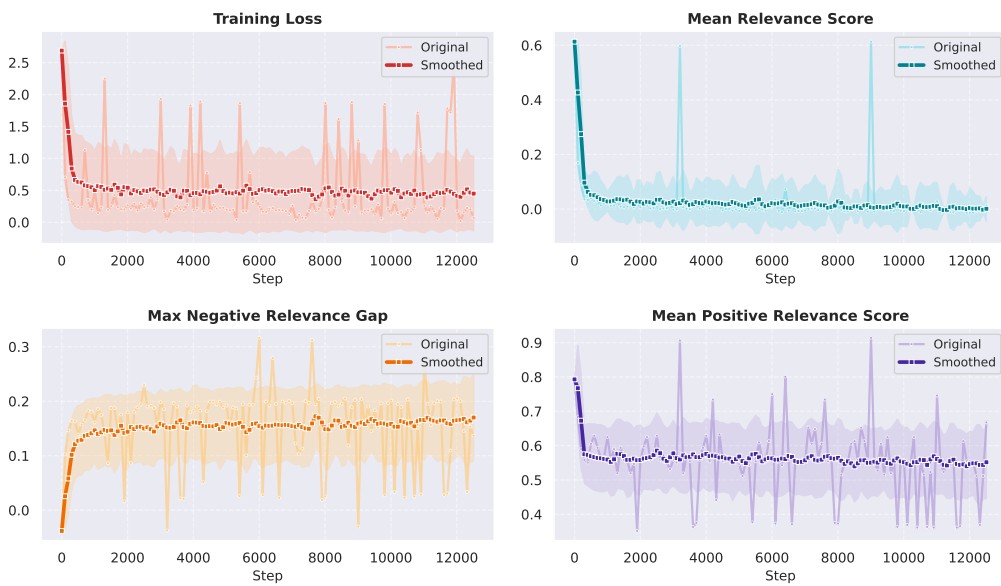

Figure 11: Training dynamics of `GVE-7B` across four metrics.

We fit a logarithmic scaling law $y = a \ln x + b$ to quantify gains per decade of data ($10\times$ increase). GVE-3B consistently shows higher relative gains: in compositional retrieval (CMP), it improves by $+0.039$ (14.7%), nearly double GVE-7B's $+0.025$ (8.7%); in coarse-grained tasks (CG), it gains $+0.057$ (11.1%), far exceeding GVE-7B's $+0.037$ (6.6%). Textual (TXT) and fine-grained (FG) retrieval follow similar trends, with GVE-3B improving by $+0.042$ (7.1%) and $+0.040$ (7.9%), respectively, versus GVE-7B's $+0.037$ (5.8%) and $+0.031$ (5.6%). Visual retrieval (VIS) scales weakly for both ($+0.024$/3.8% for GVE-3B, $+0.013$/2.1% for GVE-7B). Notably, only in long-context retrieval (LC) does GVE-7B outperform GVE-3B in both absolute ($+0.042$ vs. $+0.029$) and relative gain (5.4% vs. 3.8%).

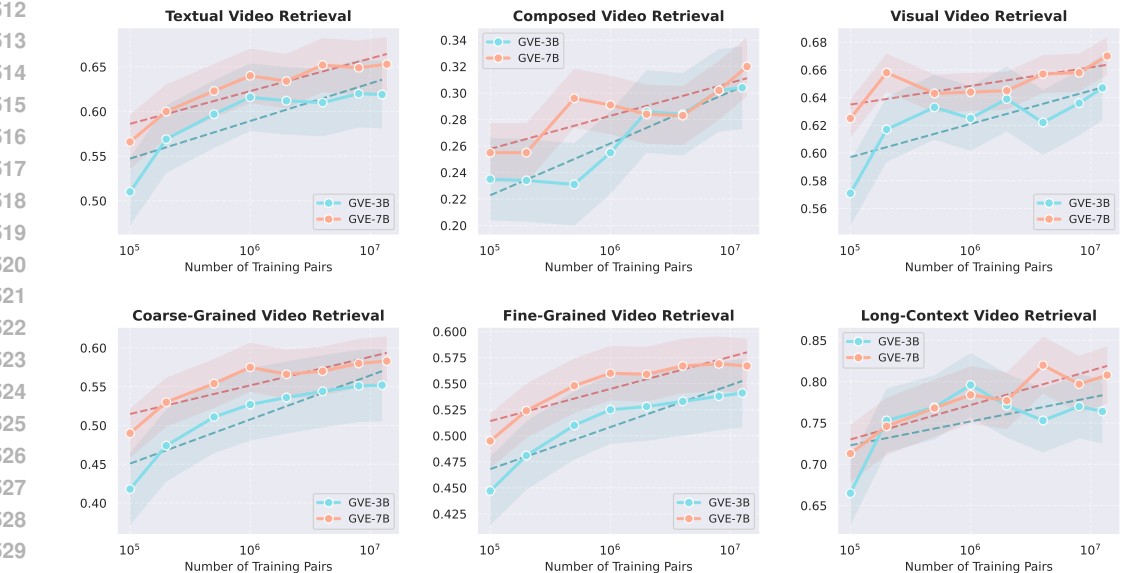

Figure 12: Performance effect from data scaling for GVE-3B and GVE-7B in detail.

This reveals a task-dependent scaling trade-off: smaller models (GVE-3B) scale more efficiently in semantic and compositional tasks, while larger models (GVE-7B) uniquely excel in long-context modeling — suggesting that model size should be chosen not only for capacity, but for alignment with the data-scaling profile of the target task.

## A.12 EXPERIMENTS OF TEST-TIME SCALING

We investigate the impact of scaling two test-time parameters: the number of sampled frames and the maximum tokens per frame (i.e., effective resolution). We analyze each parameter in isolation: when scaling the frame count from 8 to 48, the maximum token count is fixed at 200; conversely, when scaling the token count from 200 to 800, the frame count is fixed at 8.

**Temporal Scaling: the Number of Sampled Frames.** As shown in Figure 13, increasing the number of sampled frames generally improves performance. The most substantial gains appear in the Long-Context (LC) task (GVE-3B: +19.6%, GVE-7B: +12.8%), underscoring the value of denser temporal sampling for long-range reasoning. However, the Compositional (CMP) task shows a slight performance degradation, suggesting its sensitivity to potentially redundant visual cues from excessive frames. Furthermore, while GVE-7B consistently outperforms GVE-3B, the performance gap narrows as frames increase (from 0.029 to 0.024 on average scores across datasets), indicating that the larger model is more efficient at extracting information from sparser inputs.

**Spatial Scaling: the Maximum Tokens Per Frame.** In contrast, increasing the token budget yields non-monotonic returns (Figure 14). Performance for most tasks peaks around 400 tokens and then declines. This suggests that for semantic retrieval tasks, there is an optimal point of spatial detail. Scaling beyond this introduces a massive influx of fine-grained visual features that are largely redundant for the task. This over-saturation of information can dilute the effective semantic signal within the LLM's context window, making it more challenging for the model to distill the core visual concepts into the final embedding. The CMP task's performance again deteriorates with more tokens, reinforcing its sensitivity to input redundancy.

**Key Findings and Implications.** Our scaling analysis reveals four key findings:

- **Temporal Scaling (More Frames):** Provides a reliable, though diminishing, performance boost, especially for long-range tasks.

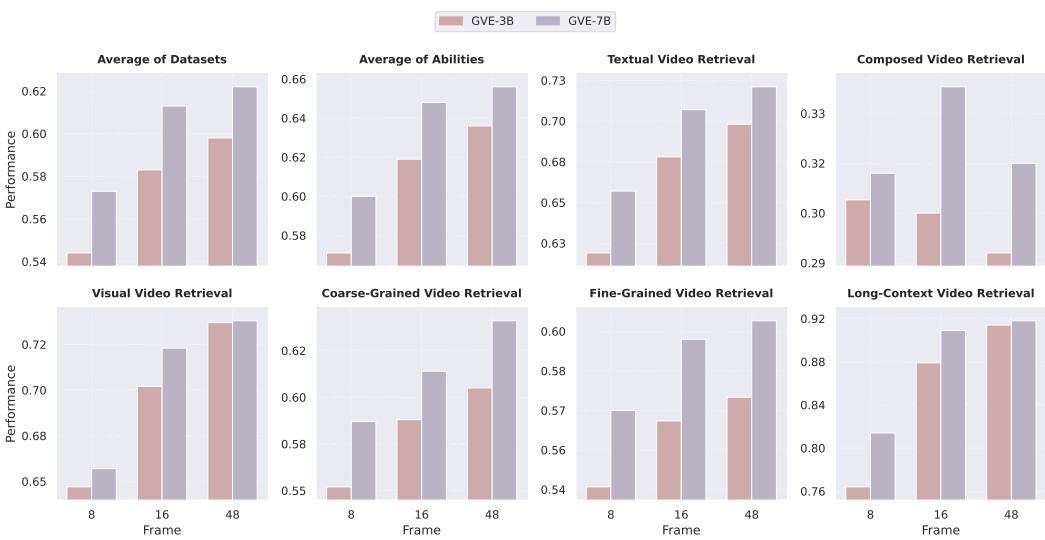

Figure 13: Model performance as a function of the number of sampled frames at inference time (max tokens fixed at 200).

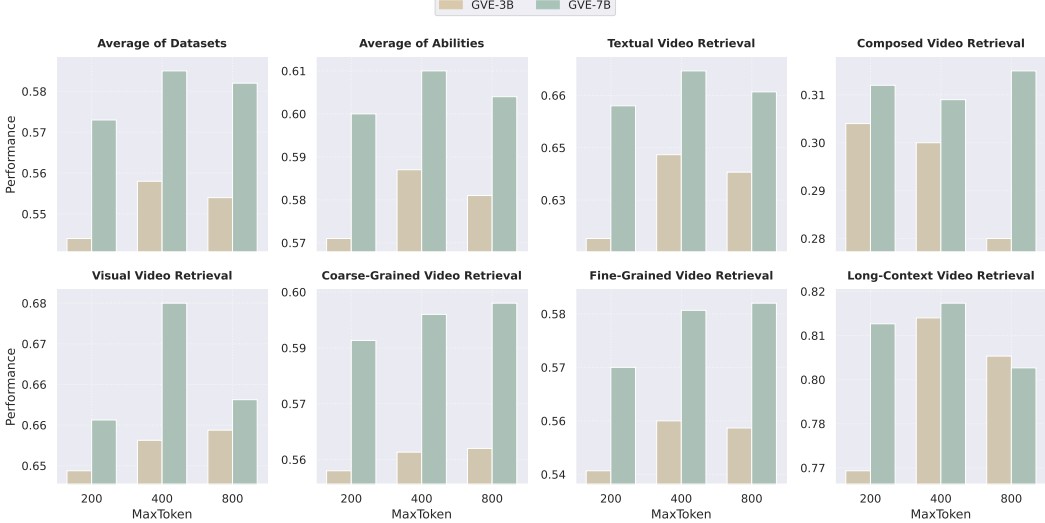

Figure 14: Model performance versus the maximum number of tokens per frame (frame count fixed at 8), determining the resolution of frames in resizing.

- **Spatial Scaling (More Tokens):** Exhibits a clear optimal point (approx. 400 tokens). Exceeding it degrades performance, indicating a trade-off between detail and distraction.
- **Model Scale Efficiency:** The larger model (GVE-7B) benefits less from input scaling, suggesting a greater intrinsic capacity to process sparse inputs effectively.

For test-time enhancement, temporal scaling is a robust strategy, whereas naive spatial scaling is not. This distinction underscores a critical insight: performance is not merely a function of total input data, but of its effective composition. It strongly motivates a shift from brute-force data scaling towards adaptive input mechanisms, such as dynamic token or frame selection, that can intelligently manage information density.

## A.13 HUMAN-IN-THE-LOOP QUALITY VALIDATION OF UVRD

To provide a ground-truth assessment of our synthesized data, we conducted a human expert evaluation on a random sample of 100 instances generated from UVRD. As referenced in Section 3.2, the

validation yielded a 95% **factual accuracy rate**. The 5 erroneous instances were categorized as follows, confirming they were minor and non-critical in nature: 2 instances with minor over/under-claims of visual details (e.g., recognize an object as a laptop/butterfly but it is not obvious); 2 instances with minor inaccuracies in camera motion description (e.g., movement hallucination); 1 instance with a subtle temporal sequencing error (e.g., reversed temporal description). This high fidelity rate validates the effectiveness of our multi-stage quality assurance process in producing reliable training data at scale.

### A.14 EXPERIMENTAL ANALYSIS OF MODALITY PYRAMID

This subsection provides detailed experimental data to quantitatively validate the design choices and overall effectiveness of the MODALITY PYRAMID, as discussed in Section 3.3 and Section 4.2.

**Empirical Validation of Task Hierarchy**  The bottom-up design of our curriculum is predicated on an assumed hierarchy of task complexity. To empirically ground this assumption, we measured the initial alignment scores of different task types before training. We computed the average cosine similarity for 100 positive pairs per task using the GME-7B (Zhang et al., 2025). The results, presented in Table 12, reveal a clear monotonic decrease in similarity as task complexity increases, from 0.68 for text-to-text (T2T) down to 0.43 for text-video-composed (TV2V) retrieval. This trend provides strong quantitative evidence for the assumed task hierarchy, justifying the curriculum's prioritization of foundational tasks.

Table 12: Average cosine similarity of positive pairs across task types at initialization. The scores reveal a natural task difficulty hierarchy, providing an empirical basis for our curriculum design. T: text, I: image, V: video.

| Task Type | T2T | T2I | T2TI | I2TI | TI2TI | T2V | TI2V | TV2V |
|---|---|---|---|---|---|---|---|---|
| Similarity | 0.68 | 0.65 | 0.59 | 0.53 | 0.52 | 0.46 | 0.45 | 0.43 |

**Impact of Temperature Scheduling**  The curriculum's dynamic task scheduling is governed by a temperature parameter $\sigma(t)$ that increases linearly over time. To analyze its effectiveness, we compared this linear schedule against a fixed, manually tuned constant temperature ($\sigma = 0.5$) for GVE-3B. As shown in Table 13, the linear schedule achieves slightly superior final performance across datasets. This result suggests that an automated, adaptive scheduling mechanism can match or exceed a carefully tuned static hyperparameter, enhancing the robustness of the training process.

Table 13: Performance comparison of temperature scheduling strategies at 3B scale.

| Scheduling Strategy | Score |
|---|---|
| Linear Schedule (Ours) | **0.571** |
| Constant Temperature ($\sigma = 0.5$) | 0.569 |

The effectiveness of this linear schedule lies in its data-driven nature. The prober model implicitly determines task difficulty based on alignment scores, leading to an adaptive schedule without requiring manual prior assumptions about which tasks are inherently "harder." We believe this data-driven approach is more robust for a universal framework than hand-crafted, task-specific schedules.

### A.15 EXPERIMENTS OF VIDEO CLASSIFICATION

We test the video embeddings for video classification by following the MMEB-V2 (Meng et al., 2025). Note that we do not train on these datasets for zero-shot evaluations, while other baselines may include them in optimization. Specifically, we evaluate video embedding models on five diverse classification benchmarks, including **Kinetics-700 (K700)** (Carreira et al., 2019), **UCF101** (Soomro et al., 2012), **HMDB51** (Kuehne et al., 2011), **SomethingSomething-V2 (SSV2)** (Goyal et al., 2017), and **Breakfast** (Kuehne et al., 2014), covering fine-grained interactions, open-domain actions, and procedural activities. Each task is cast as video-to-text retrieval over the full label set, using

standardized validation/test splits. The results are provided in Table 14. Our results show that `LanguageBind` achieves the highest mean accuracy (0.553), followed closely by `GVE-7B` (0.526) and `InternVideo2-6B` (0.526), demonstrating the efficacy of unified multimodal representations. However, performance remains suboptimal on temporally complex datasets such as Breakfast (best: 0.453) and fine-grained SSV2 (best: 0.569), revealing a critical bottleneck in modeling dynamic physical interactions.

Table 14: Video classification accuracy comparison across different models and datasets. For each column: highest score is **bolded**, second-highest is underlined.

| Model | AVG | K700 | UCF101 | HMDB51 | SSV2 | Breakfast |
|---|---|---|---|---|---|---|
| CLIP4Clip | 0.378 | 0.395 | 0.596 | 0.413 | 0.308 | 0.176 |
| InternVideo2-6B | **0.526** | **0.554** | 0.572 | 0.480 | **0.569** | **0.453** |
| VLM2Vec-V2 | 0.393 | 0.380 | 0.600 | 0.409 | 0.428 | 0.148 |
| GME-7B | 0.374 | 0.397 | 0.547 | 0.479 | 0.306 | 0.143 |
| UniME-7B | 0.306 | 0.388 | 0.377 | 0.407 | 0.190 | 0.166 |
| Unite-7B | 0.519 | 0.537 | 0.752 | **0.534** | 0.513 | 0.261 |
| GVE-3B | 0.476 | 0.489 | 0.661 | 0.483 | 0.471 | 0.277 |
| GVE-7B | **0.526** | 0.540 | **0.757** | 0.525 | 0.521 | 0.289 |

## A.16 LIMITATIONS

Our study has several practical limitations that stem from scope boundaries and experimental design.

First, all evaluations are conducted in a vision-only setting, excluding audio, transcripts, or metadata. While this aligns with standard practice in video-text retrieval, it limits applicability to real-world scenarios where multimodal cues (e.g., sound) are essential for disambiguation.

Second, inference uses a fixed protocol: videos are uniformly sampled into exactly 8 frames, and input sequences are truncated at 8192 tokens. This may disadvantage tasks requiring adaptive frame selection (e.g., sparse event detection) or longer context (e.g., hour-long videos).

Third, although UVRB covers 16 datasets across diverse tasks, it does not include specialized domains such as medical, industrial, or surveillance videos, where visual semantics and query intent differ significantly from general-domain content.

Finally, training GVE, especially the 7B variant, requires substantial computational resources, limiting accessibility for researchers with a constrained infrastructure. Efficient variants and training strategies are left for future work.

## A.17 THE USE OF LARGE LANGUAGE MODELS (LLMS)

We disclose that Large Language Models (LLMs), including Qwen3 series (Yang et al., 2025), were used as assistive tools during the preparation of this work. All output from LLMs was reviewed, revised, and validated by the authors.

In writing, we used LLMs to help correct grammatical errors and improve the phrasing of selected paragraphs for clarity and academic tone. We also consulted LLMs for LaTeX formatting advice, such as template structure, citation style, and syntax for custom environments like `promptbox`.

In coding and experiments, we used LLMs to help debug code, clarify library usage, and suggest implementation patterns. We also used LLMs to refine prompts for data synthesis, aiming to improve their interpretability by the MLLM captioner. For figure generation, we provided LLMs with rough descriptions and requested code templates; the final visualizations were manually adjusted and validated for correctness.

At no point did LLMs contribute to the formulation of research questions or scientific claims. Their role was strictly limited to assisting with language, formatting, and implementation tasks.

