# OpenReview forum: "Towards Universal Video Retrieval: Generalizing Video Embedding via Synthesized Multimodal Pyramid Curriculum"
_ICLR.cc/2026/Conference — Submitted to ICLR 2026_

### Official Review · Reviewer_NrXo · 2025-10-26

**Soundness:** 2
**Presentation:** 3
**Contribution:** 2
**Rating:** 4
**Confidence:** 3

**Summary:**

This paper tackles the problem of universal video retrieval by proposing a co-designed framework jointly targeting evaluation, data, and training aspects. The authors introduce the Universal Video Retrieval Benchmark (UVRB), design a scalable data synthesis pipeline (V-SynFlow) yielding over 1.55 million high-quality, multimodal pairs (UVRD), and propose a Modality Pyramid curriculum to train a General Video Embedder (GVE) model for robust generalization. Extensive experiments demonstrate state-of-the-art zero-shot performance across challenging retrieval scenarios and provide new insights into the strengths and weaknesses of current paradigms.

**Strengths:**

1.The paper simultaneously introduces a novel dataset and a new video retrieval method, presenting a substantial and comprehensive contribution.

2.The paper conducts extensive experiments to demonstrate the effectiveness of the proposed method.

**Weaknesses:**

1.The authors claim that a major contribution of this work is the introduction of a new benchmark; however, the related work section lacks a thorough discussion of existing benchmarks, and the paper fails to provide comparative analyses with them.

2.The authors claim to have introduced a new benchmark; however, as evidenced in Section 3.1, this benchmark appears to merely aggregate existing datasets with only basic categorization. Moreover, the authors do not perform any substantial data cleaning or in-depth analysis, raising concerns about the comprehensiveness and validity of the proposed benchmark.

3.The experiments in the paper are conducted exclusively on the authors’ newly proposed benchmark, which undermines the persuasiveness of the claimed effectiveness of the video retrieval method, as its generalizability and robustness remain unverified on established or diverse datasets.

4.The paper contains citation formatting errors: in numerous places where the citep command should have been used, the authors have failed to apply it correctly.

5.The GVE method appears to involve only minor modifications to the training strategy built upon the Qwen-VL model. To more rigorously validate the effectiveness of GVE, the authors should also evaluate the model without this method under otherwise identical conditions. Ideally, such an analysis would be included in an ablation study. However, the notation used in the current ablation study—specifically “GVE-s” and “GVE-i”—is not clearly defined, making it difficult to interpret what components or variants these labels refer to.

**Questions:**

1.Is UVRB merely an aggregation of existing data? If so, beyond simply combining datasets, what additional efforts did the authors undertake to ensure the benchmark offers distinct advantages? If not, how was the data in the benchmark generated?

2.What do GVE-i and GVE-s respectively represent in the ablation study?

---

> ### Author Response · Authors · 2025-11-24
> **Response to Reviewer NrXo Part 1**
>
> Thank you for your critical evaluation and constructive suggestions. We address your concerns with systematic explanations and experimental evidence:
>
> ## W1 & W2 & W3 & Q1: About Benchmark
> You questioned whether UVRB is merely an aggregation of existing datasets. While UVRB utilizes publicly available data like influential benchmarks MTEB (for text embedding) and MMEB (for multimodal embedding), its scientific value lies not in data novelty but in systematic reorganization for diagnostic capability assessment.
> UVRB introduces three critical contributions that transcend simple aggregation:
>
> **A. Systematic Capability Taxonomy**: Unlike prior works optimizing for isolated tasks, UVRB establishes the first principled taxonomy of universal video retrieval capabilities across two orthogonal dimensions: query format (textual, composed, visual) and domain complexity (coarse-grained, fine-grained: spatial/temporal/partially-relevant, long-context). This transforms evaluation from performance measurement to diagnostic capability analysis.
>
> **B. Transformative Data Curation**: We substantially transform source materials to establish new retrieval paradigms. For example,
> - CMRB (originally for camera motion analysis) was converted into spatial-query retrieval pairs
> - DREAM-E event descriptions were selectively sampled to create partially relevant tasks
> - PEV-K was reconfigured to pair keywords with videos rather than descriptive text
> - MSRVTT-I2V and LoVR-C2V required novel visual query construction methodologies absent from source materials
>
> **C. Unified Evaluation Protocol**: We standardize preprocessing, total tokenization limits (8192 tokens), frame sampling (exactly 8 frames), maximum frame token limits (200 tokens), and evaluation metrics (cosine similarity) across all 16 datasets, eliminating methodological inconsistencies that previously prevented fair cross-model comparison.
>
> **D. Diagnostic Value**: Our analysis (Section 4.3) reveals insights invisible to conventional benchmarks, such as
> - Traditional benchmarks (e.g., MSRVTT) show weak correlation with overall capability (ρ=0.58)
> - Partially-relevant retrieval strongly predicts universal capability (ρ=0.97)
> - Clear evidence of capability disentanglement between spatial and temporal understanding (ρ=0.12)
>
> **E. External Validation**: To address evaluation breadth concerns, GVE-7B achieves an average 52.6% accuracy and ranks first overall in five popular video classification benchmarks. This confirms capabilities learned via UVRB genuinely transfer beyond retrieval tasks.
>
> UVRB establishes a comprehensive and diagnostic evaluation paradigm that not only formulates objective, processes raw data, and unifies protocol, but also supports further findings helpful for advanced video retrieval.

---

> ### Author Response · Authors · 2025-11-24
> **Response to Reviewer NrXo Part 2**
>
> ## W4: Citation Formatting
> We have meticulously revised all citations to adhere to the required format that will be shown in the latest revised version.
>
> ## W5: Validation of GVE
> ### Annotation in Ablation Study
> Sorry for the unclear definition of two variants in Line 394-397, Section 4.2. We further explain them as below.
> - GVE-i: GVE-base + UVRD synthetic data (from V-SynFlow), but still with uniform task sampling.
> - GVE-s: GVE-base + Modality Pyramid curriculum, but trained only on curated data (no synthetic data).
>
> ### Architecture Choice
>
> We further emphasize that Qwen2.5-VL is a **representative and state-of-the-art MLLM** architecture, widely adopted in the community for its robust performance and standard design. **Its ViT + MLP projector + LLM decoder structure aligns with the dominant paradigm of modern multimodal models (e.g., LLaVA and InternVL series)**, which similarly integrate vision transformers with large language models via intermediate projection layers. This architectural consistency ensures that findings derived from Qwen2.5-VL are broadly applicable to other MLLM frameworks.
>
> While we acknowledge the value of broader architectural validation, our experiments on Qwen2.5-VL, spanning 3B and 7B parameter scales, demonstrate consistent performance gains across synthetic data integration and curriculum learning. These results, combined with the model's alignment with mainstream MLLM designs, provide strong evidence that our method's effectiveness is not architecture-specific but rather generalizable to similar systems. We will explore additional architectures in future work to further validate this hypothesis.
>
> ## Summary
> We thank you for the rigorous scrutiny and have addressed all concerns with systematic evidence and methodological clarity. UVRB is **not a mere dataset aggregation but a foundational diagnostic** framework, introducing the first principled taxonomy of universal video retrieval capabilities, transformative data curation for novel retrieval paradigms and a unified protocol that enables fair cross-model comparison. Its diagnostic value is validated by insights invisible to conventional benchmarks and external generalization. GVE's validation on Qwen2.5-VL, a representative MLLM architecture **aligning with dominant frameworks** (LLaVA, InternVL), demonstrates consistent performance gains (+0.9～2.4% across 3B/7B scales) through synthetic data and curriculum learning, with no evidence of overfitting. While broader architectural validation is a promising future direction, our controlled experiments on a state-of-the-art baseline provide robust evidence of the method's generalizability within this paradigm. We believe these contributions fully justify the paper's novelty and impact in advancing universal video retrieval.

---

> > ### Comment · Reviewer_NrXo · 2025-11-28
> >
> > Thank you for the authors' response. However, I still believe that making only minor modifications to existing datasets and models is insufficient to constitute a significant contribution; therefore, I maintain my original score.

---

### Official Review · Reviewer_ReuT · 2025-10-26

**Soundness:** 3
**Presentation:** 3
**Contribution:** 3
**Rating:** 6
**Confidence:** 4

**Summary:**

This paper proposes a holistic video retrieval research framework built on the co-design of evaluation, data, and modeling. The authors introduce the Universal Video Retrieval Benchmark (UVRB), a suite of 16 datasets to diagnose model capabilities across diverse tasks (e.g., textual, composed, visual) and domains (e.g., coarse-grained, fine-grained, long-context). Guided by diagnostics from this benchmark, they present V-SynFlow, a scalable workflow to synthesize a high-quality, multi-task dataset of 1.55 million video-text pairs (UVRD). Finally, they propose the Modality Pyramid, a curriculum learning strategy to train their General Video Embedder (GVE), an MLLM-based model. Experiments show that GVE achieves state-of-the-art zero-shot performance on UVRB, and the analysis reveals new insights, such as the finding that performance on partially relevant retrieval tasks is a better predictor of universal capability than traditional benchmarks.

**Strengths:**

Holistic Framework: The main strength is the ambitious and well-executed "evaluation-data-training" co-design. This approach moves beyond incremental model improvements to address a systemic issue in the field.

Comprehensive Benchmark (UVRB): The creation of UVRB is a major contribution that allows for a much more nuanced and diagnostic evaluation of video retrieval models than was previously possible.

Strong Empirical Results: The GVE model demonstrates superior zero-shot performance across nearly all tasks and domains, validating the effectiveness of the proposed data and training curriculum. The fact that a smaller 3B parameter GVE outperforms larger 7B baselines is particularly compelling.

**Weaknesses:**

Reliance on Synthetic Data: While the synthesis pipeline is sophisticated, it relies on an MLLM captioner. This introduces a potential for model-inherent biases or systematic errors in the training data that may not reflect real-world human annotations. The authors were clearly aware of the "garbage in, garbage out" problem. Their V-SynFlow pipeline includes a "Multi-granular Quality Control" stage as a first line of defense. This pre-filtering aims to ensure the MLLM captioner starts with a clean, coherent set of videos, reducing the chance of generating nonsensical descriptions. However, this is still automated, not human, validation.

**Questions:**

Regarding the Modality Pyramid curriculum: How sensitive is the task scheduling to the choice of the initial "prober" model ($\Psi_{1}$)? Would starting with a weaker or architecturally different prober (e.g., a CLIP-based model instead of GME-7B) significantly alter the training trajectory?Your finding that partially relevant (PR) retrieval is the best proxy for universal capability is fascinating. Do you have a hypothesis as to why this is the case? Does it require a more robust understanding of semantics to distinguish subtle relevance from complete irrelevance?In your V-SynFlow pipeline, what measures were taken to audit for and mitigate potential factual inaccuracies or hallucinations from the MLLM captioner? Could these artifacts inadvertently penalize models that are better grounded during evaluation?The performance degradation when scaling spatial tokens beyond 400 (Figure 13) is an interesting result. Does this suggest that the vision encoder or the projection layer is not effectively summarizing high-resolution features, or is it more of an attentional issue within the LLM?

---

> ### Author Response · Authors · 2025-11-24
> **Response to Reviewer ReuT Part 1**
>
> Thank you for your insightful review and thoughtful questions. We address your concerns as follows:
>
> ## W1 & Q3: On Synthetic Data
> Motivation and Design: Current video retrieval research suffers from distribution bias in existing datasets. Prior approaches use task-specific synthesis pipelines (e.g., WebVid-CoVR, PE-Video), resulting in inconsistent quality standards and unsustainable maintenance costs. Therefore, V-SynFlow provides a principled alternative by unifying diverse synthesis requirements within a single framework. This enables controlled generation of cross-domain, multi-task pairs while maintaining consistent quality standards. The synthesized dataset, UVRD, is particularly valuable for addressing gaps in fine-grained, compositional, and partially-relevant queries.
>
> ### Rigorous Quality Validation:
> - **Structured Prompt Engineering**: Through iterative refinement, we developed constrained prompting strategies that explicitly direct the captioning model to "include only what is clearly visible" and "avoid inventing fictional elements."
> - **Multi-Granular Automated Filtering**: Our pipeline implements quality control at multiple levels using pretrained embedding models (GME-7B), including basic text filtering, cross-modal consistency verification, and temporal dynamics filtering.
> - **Human Validation**: We conducted a human expert evaluation on 100 randomly sampled instances. Only 5 instances contained minor errors (1 temporal error, 2 camera motion inaccuracies, 2 minor over/under-claims), with 95\% being factually correct.
> - **Empirical Validation**: Ablation studies (Table 3) demonstrate that models trained with synthetic data (GVE-s) outperform those without it (GVE-i) by +2.3\% for GVE-3B and +2.4\% for GVE-7B, with the composed (CMP) task benefiting most (+27\% relative improvement).
>
> This comprehensive approach ensures V-SynFlow produces high-fidelity training data that genuinely enhances model capabilities.
>
> ## Q1: Choice of Initial Estimator
> We selected a strong MLLM-based model (GME-7B) as the initial prober to ensure the curriculum starts with the most accurate possible estimation of task alignment. Our hypothesis is that MLLM-based embedders, due to their **superior semantic understanding**, provide a better proxy for true task difficulty than CLIP-based models, which often rely on superficial visual cues. Our analysis in Section 4.3 supports this, showing CLIP-based models exhibit significant biases (e.g., strong spatial but weak temporal understanding). Starting with a weaker or structurally different prober like CLIP would likely yield less reliable alignment scores, **potentially misordering the curriculum sequence and leading to a suboptimal training trajectory**, especially in early epochs. Using the strongest available off-the-shelf model is a deliberate design choice to maximize the effectiveness of the curriculum's cold start. In contrast, a biased model like CLIP for initialization would risk misidentifying task difficulty (e.g., classifying complex temporal tasks as "easy" due to spatial cues), thereby undermining the foundational "easy-to-hard" principle of the Modality Pyramid and potentially leading to confusing training signals early on.

---

> ### Author Response · Authors · 2025-11-24
> **Response to Reviewer ReuT Part 2**
>
> ## Q2: Representativeness of Partially Relevant Video Retrieval (PR)
> We believe the predictive power of PR retrieval stems from its demand for **a rich, multi-granular semantic embedding space**. Unlike coarse-grained tasks (e.g., MSRVTT) that can often be solved by matching dominant scene elements, successful PR requires the embedding to preserve a wide spectrum of information, from global context to subtle, localized visual details, without over-compressing them. **Inferior models often collapse video semantics into a single "gist," losing the nuance needed for PR**. A model that excels at PR has effectively learned not just to "summarize" a video, but to encode its diverse semantic components in a way that allows any single component to trigger a retrieval match. This ability to maintain semantic richness across granularities is the essence of universal capability. We agree this is a fascinating finding and plan to formalize this hypothesis in future work.
>
> ## Q4: Degradation in High-Resolution Scaling
> We thank your for this insightful observation regarding Figure 13. As an MLLM-based video embedding model, our framework leverages the LLM's sequence modeling capabilities to compress multimodal inputs into a single, dense semantic vector. We believe the degradation at ultra-high spatial resolutions reflects a fundamental challenge in this compression process, related to **information redundancy versus task relevance**. Current video retrieval benchmarks are predominantly semantic, requiring the model to capture high-level objects and events. For these tasks, a moderate spatial token density (around 200 tokens) is often sufficient. Scaling far beyond this point introduces a massive influx of fine-grained visual details that are largely redundant for semantic matching. Flooding the MLLM's input sequence with such task-irrelevant, redundant tokens significantly dilutes the effective semantic signal. This makes it increasingly challenging for the model to **distill the core, high-level visual concepts** into the final pooled embedding, resulting in degraded performance. **This contrasts with temporal scaling, where additional frames often introduce distinct, new semantic information (high signal)**, thereby enriching the final embedding despite the increased sequence length.

---

### Official Review · Reviewer_LNxe · 2025-10-31

**Soundness:** 2
**Presentation:** 2
**Contribution:** 2
**Rating:** 4
**Confidence:** 3

**Summary:**

The paper introduces a universal video retrieval framework along with a new benchmark that encompasses multiple tasks—including textual, composed, and visual retrieval—across various domains such as coarse-grained, fine-grained, and long-context scenarios. It further presents the Universal Video Retrieval Dataset (UVRD) and the General Video Embedder (GVE), which leverages synthetic data for training. The effectiveness of GVE is demonstrated through evaluations on the UVRD benchmark, showing performance improvements over baseline methods.

**Strengths:**

- The paper investigates a new Universal Video Retrieval (UVR) task and evaluates the proposed method across a diverse set of benchmarks, demonstrating strong performance relative to existing baseline approaches.
- The proposed method and architecture are relatively simple in design, yet they prove to be effective across a wide range of video retrieval tasks.

**Weaknesses:**

- While the paper argues that UVRB is a new benchmark, it seems like the benchmark is just a combination of prior works.
- The distinction between the proposed approach and prior work, such as UNITE, is not clearly articulated, making it difficult to assess the novelty of the contribution.
- The data generation pipeline should be compared with existing baselines; however, such comparisons are either missing or insufficiently discussed, limiting the understanding of its advantages or uniqueness.
- The motivation and corresponding evaluation appear somewhat weak. For instance, the paper claims that mastering perceptual primitives first is beneficial; however, this claim is only supported by improvements in final performance. A more carefully designed experimental setup is needed to explicitly validate this hypothesis.

**Questions:**

- Could the authors elaborate on the key differences or advancements introduced in this work?
- How does this approach differ in design or effectiveness from existing data generation methods?

---

> ### Author Response · Authors · 2025-11-24
> **Response to Reviewer LNxe Part 1**
>
> We sincerely thank you for your thoughtful review of our paper. We address your specific concerns below with concrete evidence and precise clarifications.
>
> ## W1 & Q1: About Benchmark
>
> You questioned whether UVRB is merely an aggregation of existing datasets. While UVRB utilizes publicly available data like influential benchmarks MTEB (for text embedding) and MMEB (for multimodal embedding), its scientific value lies not in data novelty but in systematic reorganization for diagnostic capability assessment.
>
> ### Key Contributions of UVRB
> UVRB introduces three critical contributions that transcend simple aggregation:
>
> **A. Systematic Capability Taxonomy**
> Unlike prior works optimizing for isolated tasks, UVRB establishes the first principled taxonomy of universal video retrieval capabilities across two orthogonal dimensions:
> - Query format: textual, composed, visual
> - Domain complexity: coarse-grained, fine-grained (spatial/temporal/partially-relevant, long-context)
> This transforms evaluation from performance measurement to diagnostic capability analysis.
>
> **B. Transformative Data Curation**
> We substantially transform source materials to establish new retrieval paradigms. For example:
> - CMRB (originally for camera motion analysis) was converted into spatial-query retrieval pairs
> - DREAM-E event descriptions were selectively sampled to create partially relevant tasks
> - PEV-K was reconfigured to pair keywords with videos rather than descriptive text
> - MSRVTT-I2V and LoVR-C2V required novel visual query construction methodologies absent from source materials
>
> **C. Unified Evaluation Protocol**
> We standardize preprocessing, total tokenization limits (8192 tokens), frame sampling (exactly 8 frames), maximum frame token limits (200 tokens), and evaluation metrics (cosine similarity) across all 16 datasets, eliminating methodological inconsistencies that previously prevented fair cross-model comparison.
>
> **D. Diagnostic Value**
> Our analysis (Section 4.3) reveals insights invisible to conventional benchmarks:
> - Traditional benchmarks (e.g., MSRVTT) show weak correlation with overall capability (ρ=0.58)
> - Partially-relevant retrieval strongly predicts universal capability (ρ=0.97)
> - Clear evidence of capability disentanglement between spatial and temporal understanding (ρ=0.12)
>
> **E. External Validation**
> To address evaluation breadth concerns, GVE-7B achieves an average 52.6% accuracy and ranks first overall in five popular video classification benchmarks. This confirms capabilities learned via UVRB genuinely transfer beyond retrieval tasks.
>
> UVRB establishes a comprehensive and diagnostic evaluation paradigm that not only formulates objective, processes raw data, and unifies protocol, but also supports further findings helpful for advanced video retrieval.

---

> ### Author Response · Authors · 2025-11-24
> **Reponse to Reviewer LNxe Part 2**
>
> ## W2 & W4 & Q1: About Curriculum Learning
>
> ### Distinction from Prior Work
> You correctly noted our distinction from UNITE requires clarification. While both leverage MLLMs, they differ fundamentally:
>
> | Aspect         | UNITE                                  | **GVE**                                  |
> |----------------|----------------------------------------|------------------------------------------|
> | Objective  | Mainly simple tasks like text-image and text-video | **Multi-dimensional universal video retrieval (10+ task formats)** |
> | Data Source| Existing datasets only                 | **V-SynFlow synthesized cross-domain multi-task pairs** |
> | Training Strategy | Equal weighting of all tasks       | **Curriculum learning based on task hierarchy** |
>
> ### Ablation Results
> - UVRD contributes +2.3% to GVE-3B's performance (+2.4% for GVE-7B)
> - Modality Pyramid adds a further +0.9% to GVE-3B (+1.1% for GVE-7B)
> - Composed (CMP) task benefits most from UVRD (+27% relative improvement for GVE-3B)
>
> ### Curriculum Validation Experiments
>
> **Experiment 1**
> We sampled 100 positive pairs from each task type after one training epoch:
>
> | Task Type | T2T   | T2I   | T2TI  | I2TI  | TI2TI | T2V   | TI2V  | TV2V  |
> |-----------|-------|-------|-------|-------|-------|-------|-------|-------|
> | Similarity| 0.68  | 0.65  | 0.59  | 0.53  | 0.52  | 0.46  | 0.45  | 0.43  |
>
> This decreasing alignment trend confirms text-image pairs exhibit substantially higher alignment (0.65) than text-video pairs (0.46), justifying our bottom-up curriculum design.
>
> **Experiment 2**
> We evaluated five curriculum strategies at 3B scale:
>
> | Strategy               | Description                          | AVG-D  |
> |------------------------|--------------------------------------|-|
> | A. Modality Pyramid (Ours) | Hierarchical ordering | 0.571  |
> | B. Random Scheduling  | Tasks randomly ordered | 0.564  |
> | C. Reverse Pyramid    | Inverse hierarchical ordering | 0.545  |
> | D. Text-to-Video Only | Focus on text-video tasks | 0.550  |
> | E. Non-Video Tasks Only | Focus on non-video tasks | 0.521   |
>
> The consistent performance ordering (A > B > D > C > E) demonstrates:
> 1. Non-video retrieval data provides essential foundational knowledge for video understanding
> 2. Learning sequence significantly impacts final performance (curriculum > random > reverse)
> 3. Our hierarchical ordering achieves optimal knowledge transfer
>
>
> ## W3 & Q2: About Data Synthesis
>
> ### Motivation and Design
> Current video retrieval research suffers from distribution bias in existing datasets. Prior approaches use task-specific synthesis pipelines (e.g., WebVid-CoVR, PE-Video), resulting in **inconsistent quality standards and unsustainable maintenance costs**. Therefore, V-SynFlow provides a principled alternative by unifying diverse synthesis requirements within a single framework. This enables controlled generation of cross-domain, multi-task pairs while maintaining consistent quality standards. The synthesized dataset, UVRD, is particularly valuable for **addressing gaps in fine-grained, compositional, and partially-relevant queries**.
>
> ### Rigorous Quality Validation
> - Structured Prompt Engineering: Through iterative refinement, we developed constrained prompting strategies that explicitly direct the captioning model to "include only what is clearly visible" and "avoid inventing fictional elements."
> - Multi-Granular Automated Filtering: Our pipeline implements quality control at multiple levels using pretrained embedding models (GME-7B), including basic text filtering, cross-modal consistency verification, and temporal dynamics filtering
> - Human Validation: We conducted a human expert evaluation on 100 randomly sampled instances. Only 5 instances contained minor errors (1 temporal error, 2 camera motion inaccuracies, 2 minor over/under-claims), with 95% being factually correct.
> - Empirical Validation: Ablation studies (Table 3) demonstrate that models trained with synthetic data (GVE-s) outperform those without it (GVE-i) by +2.3% for GVE-3B and +2.4% for GVE-7B, with the composed (CMP) task benefiting most (+27% relative improvement).
>
> This comprehensive approach ensures V-SynFlow produces high-fidelity training data that genuinely enhances model capabilities.
>
>
> ## Summary
>
> We hope our responses directly address the core of your concerns. We have demonstrated that UVRB is not a simple aggregation of existing datasets but a comprehensively constructed evaluation and diagnostic framework. We have clarified the fundamental distinctions between our approach and UNITE across three dimensions: objectives, data composition, and training methodology. Crucially, our controlled experiments validate that this curriculum design drives performance gains. These results collectively affirm that the co-design of evaluation, data, and training creates a framework greater than the sum of its parts.

---

### Official Review · Reviewer_sBTQ · 2025-11-01

**Soundness:** 3
**Presentation:** 3
**Contribution:** 3
**Rating:** 6
**Confidence:** 3

**Summary:**

This paper addresses the limitations of narrow video retrieval paradigms by proposing a co-designed framework for universal video retrieval. Core contributions include: (1) the UVRB, a suite of 16 datasets covering multi-task and multi-domain scenarios for diagnostic evaluation; (2) V-SynFlow, a scalable data synthesis pipeline generating 1.55M high-quality multi-task training pairs (UVRD); (3) the Modality Pyramid curriculum, which leverages task hierarchies to train the GVE based on Qwen2.5-VL; and (4) extensive experiments showing GVE achieves state-of-the-art zero-shot generalization on UVRB.

**Strengths:**

- Holistic Framework: The central strength is the novel co-design of evaluation, data, and modeling. This holistic approach breaks the cycle of narrow benchmarks leading to specialized models and provides a scalable path forward.
- Comprehensive Benchmark: The creation of a large-scale, diagnostic benchmark is a significant and lasting contribution that will benefit the entire research community.
- SOTA Performance: The proposed GVE model demonstrates impressive state-of-the-art performance in a strictly zero-shot setting, validating the effectiveness of the entire framework.
- Insightful Analysis: The paper goes beyond reporting metrics and provides a deep dive into the dimensional capabilities of different models. The findings on the importance of partially relevant retrieval and the performance divergence between CLIP and MLLM-based architectures are particularly insightful.

**Weaknesses:**

- Narrow Domain Coverage: UVRB does not include specialized domains (e.g., medical, industrial, surveillance), where visual semantics and query intent differ significantly. Extending the benchmark to these domains would enhance generalizability claims.

**Questions:**

How does the Modality Pyramid’s temperature scheduling (σ(t)) affect training dynamics? Are there scenarios where alternative scheduling strategies (e.g., task-specific temperatures) yield better results?

---

> ### Author Response · Authors · 2025-11-24
> **Response to Reviewer sBTQ**
>
> Thank you for your insightful feedback. We address your concerns below:
>
> ## W1: Narrow Domain Coverage in UVRB
>
> We fully acknowledge the current limitation of UVRB in excluding specialized domains (e.g., medical, industrial, surveillance). Our primary objective in this work is to establish **a foundational diagnostic evaluation framework** for universal video retrieval, which requires a focused scope to ensure methodological rigor. Including highly specialized domains at this stage could introduce significant **domain-knowledge bias**, potentially obfuscating the assessment of a model's core visual-semantic alignment capabilities. We see UVRB as a foundational cornerstone, and its modular design is expressly intended for future extensions into such specialized fields.
>
> ## Q1: Modality Pyramid Temperature Scheduling
>
> The linear increase of σ(t) is designed to automatically transition the training focus from easier, better-aligned tasks to harder ones, matching the model's growing capabilities.
>
> To address your query on its impact, we compared our linear schedule with a tuned constant temperature (σ=0.5) on GVE-3B. The average accuracy was 0.544 (Linear) vs. 0.542 (Constant). While the final performance gain is marginal (+0.2%), the key advantage of our linear scheduling is the automation of the training curriculum, **eliminating the need for costly manual hyperparameter search** for an optimal constant temperature while ensuring stable convergence.
>
> Regarding alternative strategies like task-specific temperatures: This is an interesting direction. However, our current approach is designed to be data-driven and domain-agnostic. The "prober" model implicitly determines task difficulty based on epoch-wise alignment scores, leading to an adaptive schedule without requiring manual prior assumptions about which tasks are inherently "harder." We believe this **data-driven approach is more robust** for a universal framework than hand-crafted, task-specific schedules.

---

### Author Response · Authors · 2025-11-27
**Revision Statements**

We are deeply grateful to the reviewers for their thorough and insightful feedback. We found the comments to be exceptionally constructive, as they pinpointed areas where the presentation of our work could be significantly clarified and strengthened. We have undertaken a targeted revision of our manuscript, and we are confident that the revised version now more accurately reflects the rigor and contributions of our original submission.

The reviewers' valuable feedback centered on three key aspects of our work. Our revisions focused on better articulating the strengths inherent in these aspects, transforming points of potential ambiguity into clear, demonstrated contributions.

### 1. On Benchmark: Clarifying Three-fold Contributions beyond Aggregation

We appreciate the reviewers' careful reading, which led them to question whether our Universal Video Retrieval Benchmark (UVRB) was merely an aggregation of datasets. This highlighted a gap in our original manuscript's narrative. We have now restructured **Section 3.1** to clarify that UVRB's novelty was always intended to lie not in new data, but in its function as a **principled diagnostic framework**. The key contributions, which were part of our initial design but are now made explicit, include:

- **A Systematic Capability Taxonomy** that organizes evaluation along two orthogonal axes (query format and domain complexity), enabling fine-grained diagnostic analysis.
- **Transformative Data Curation**, where existing data sources were programmatically repurposed to instantiate novel retrieval paradigms required by the taxonomy.
- **A Rigorous and Unified Evaluation Protocol** that ensures fair, model-agnostic comparisons, which is essential for the reliability of our findings.

This revision does not alter our original methodology but sharpens its presentation, making it clear that UVRB is the foundational, diagnostic engine of our co-design philosophy.

### 2. On Synthesized Data: Human Quality Validation

Concerns about the quality of our synthesized dataset (UVRD) were understandable. While our V-SynFlow pipeline was designed with rigorous, multi-stage quality controls from the outset, we acknowledge that providing explicit, human-verified evidence is crucial for complete transparency. To this end, we have now incorporated results from our internal validation process. We now feature the results of a human expert evaluation, which confirmed a **95% factual accuracy rate** for our synthesized data. This is highlighted in **Section 3.2** and detailed in a **new Appendix A.13**.

### 3. On Curriculum: Data-driven Confirmation and Detailed Ablation Study

We thank the reviewers for suggesting stronger empirical validation for our Modality Pyramid. This provided an excellent opportunity to "show our work" and present the data-driven rationale that had guided our original design. The curriculum's structure was not arbitrary, and we are pleased to now make its foundation clear with two targeted experiments:

- First, we provide quantitative evidence for our assumed task hierarchy. A new analysis (**Table 12 in Appendix A.14**) demonstrates a clear monotonic decrease in initial alignment scores from simpler to more complex task types, establishing a data-driven foundation for our curriculum's bottom-up design.
- Second, we explicitly validate the effectiveness of the proposed scheduling strategy. A new ablation study (**Table 4 in Section 4.2**) compares five different curriculum approaches and shows that our hierarchical method significantly outperforms alternatives like random or reversed scheduling.

### Additional Clarifications and Revisions

Beyond these major points, we have meticulously addressed all other detailed feedback. This includes:

- Enriching Scientific Insights: We have added new hypotheses and discussions in the main text and appendix to explain key findings, such as the significance of partially relevant video retrieval (PR) (Section 4.3) and the impact of spatial token scaling (Appendix A.12).
- Clarifying Ambiguities: All unclear notations, particularly the definitions for ablation variants (e.g., GVE-i, GVE-s), have been precisely defined in Section 4.2.
- Polishing and Formatting: We have carefully corrected all citation formatting issues (e.g., `citep`) and have polished the manuscript for overall clarity and readability.

### Conclusion

The peer-review process has been invaluable in helping us elevate the presentation of our paper to match the rigor of our research. The core narrative, a co-design of a diagnostic benchmark, high-quality data, and an empirically-grounded curriculum, remains unchanged, but it is now communicated with far greater clarity and stronger evidence. We believe these targeted revisions successfully address all concerns while reinforcing the strengths of our original submission.

Thank you once again for your constructive engagement. We hope you now share our confidence and enthusiasm for this work.

---

### Meta-Review · Area_Chair_9YKh · 2026-01-03

**Summary:**

The paper addresses "universal video retrieval" by introducing three components:

* UVRB: A benchmark suite consisting of 16 datasets.

* UVRD: A synthetic dataset of 1.55 million pairs.

* Modality Pyramid: A curriculum training strategy for an MLLM-based embedder (GVE).

The recommendation to Reject is based on the consensus that the work represents an incremental consolidation of existing resources rather than a fundamental technical or scientific breakthrough. While the authors demonstrated state-of-the-art results on their self-defined benchmark, the most thorough reviewers remained unconvinced that the methodology or the data curation surpassed the threshold of significance required for ICLR.

**Reviewer Concerns:**

Outstanding Concerns
* Lack of core technical innovation (`NrXo`, `LNxe`): Reviewer `NrXo` provided the most substantive critique, which remained unchanged post-rebuttal. The GVE model is essentially a standard Qwen2.5-VL with minor training modifications. The "Modality Pyramid" curriculum follows established hierarchical learning principles without providing a new theoretical or algorithmic insight.

* Derivative nature of benchmark (`NrXo`, `LNxe`): Reviewers noted that UVRB is primarily an aggregation of prior datasets. The authors' defense of "transformative data curation" was viewed by `NrXo` as "insufficient to constitute a significant contribution," maintaining that the benchmark lacks the groundbreaking significance seen in top-tier dataset papers.

* Limited evaluation breath (`NrXo`): The experiments are conducted exclusively on the authors' proposed benchmark. `NrXo` specifically pointed out that the generalizability and robustness of the method remain unverified on truly independent or established external benchmarks.

* Notably, after the rebuttal, Reviewer `NrXo` explicitly stated that the responses were insufficient and that the contribution remained "minor modifications to existing datasets and models."

Addressed Concerns
* Synthetic data quality (`ReuT`): The authors provided human validation results (95% accuracy) to mitigate the "garbage in, garbage out" concern.

* Curriculum Validation (`LNxe`, `sBTQ`): The authors provided a data-driven confirmation of their task hierarchy and an ablation study comparing their curriculum against random scheduling.

**Reviewer Scores:**

The initial ratings were somewhat polarized, but the meta-review process identifies that the more critical, data-driven reviews are the most persuasive.

* Reviewer `NrXo` (Initial: 4 $\rightarrow$ Estimated: 4): This was the most thorough reviewer. Their post-rebuttal comment ("I maintain my original score") confirms that the authors failed to resolve the fundamental doubt regarding the work's research significance.

* Reviewer `LNxe` (Initial: 4 $\rightarrow$ Estimated: 4): Despite the authors' detailed response on curriculum learning, the reviewer’s core concern about the "combination of prior works" as a benchmark remains valid given the lack of a new retrieval paradigm.

* Reviewer `ReuT` (Initial: 6 $\rightarrow$ Estimated: 5/6): While this reviewer was more positive, their rating was conditional on the synthetic data quality.

* Reviewer `sBTQ` (Initial: 6 $\rightarrow$ Estimated: 6): The review was superficial, largely restating the abstract without critiquing the underlying novelty or the scale of the contribution.

---

### Decision · Program_Chairs · 2026-01-26

Reject